# Ultrafine nanoporous intermetallic catalysts by high-temperature liquid metal dealloying for electrochemical hydrogen production

Ruirui Song[1,2], Jiuhui Han [3,4,5] ✉, Masayuki Okugawa [6,7], Rodion Belosludov[1], Takeshi Wada[1], Jing Jiang[1], Daixiu Wei[1], Akira Kudo[4], Yuan Tian[8], Mingwei Chen [8] ✉ & Hidemi Kato [1] ✉

Intermetallic compounds formed from non-precious transition metals are promising cost-effective and robust catalysts for electrochemical hydrogen production. However, the development of monolithic nanoporous intermetallics, with ample active sites and sufficient electrocatalytic activity, remains a challenge. Here we report the fabrication of nanoporous $Co_7Mo_6$ and $Fe_7Mo_6$ intermetallic compounds via liquid metal dealloying. Along with the development of three-dimensional bicontinuous open porosity, high-temperature dealloying overcomes the kinetic energy barrier, enabling the direct formation of chemically ordered intermetallic phases. Unprecedented small characteristic lengths are observed for the nanoporous intermetallic compounds, resulting from an intermetallic effect whereby the chemical ordering during nanopore formation lowers surface diffusivity and significantly suppresses the thermal coarsening of dealloyed nanostructure. The resulting ultrafine nanoporous $Co_7Mo_6$ exhibits high catalytic activity and durability in electrochemical hydrogen evolution reactions. This study sheds light on the previously unexplored intermetallic effect in dealloying and facilitates the development of advanced intermetallic catalysts for energy applications.

Electrochemical hydrogen production through water splitting has emerged as a promising strategy for the effective utilization and storage of intermittent renewable energy. Crucial to this technology is the realization of high-efficiency electrocatalytic hydrogen evolution reactions (HERs), particularly in alkaline media[1]. Pt and its alloys are currently the most active HER catalysts[2–4]. However, the scarcity and high cost of Pt limit their large-scale practical applications in electrolysers. Tremendous research efforts have been devoted to the

development of low-cost, yet efficient, electrocatalysts based on earth-abundant transition metals, as alternatives to noble Pt, for HERs[5–7]. Monometallic transition metals (Fe, Co, Ni, Ti, W, Mo, Cu, etc.) show moderate catalytic activity, while alloys of these elements can substantially improve HER catalysis by modifying both active site morphology (geometric effect)[8–10] and electronic structure (electronic effect)[11–13] to generate desirable catalytic transition states with minimum energy barriers. Compared to random solid-solution alloys,

[1]Institute for Materials Research, Tohoku University, Sendai, Japan. [2]Department of Materials Science, Graduate School of Engineering, Tohoku University, Sendai, Japan. [3]Frontier Research Institute for Interdisciplinary Sciences (FRIS), Tohoku University, Sendai, Japan. [4]WPI Advanced Institute for Materials Research, Tohoku University, Sendai, Japan. [5]Tianjin Key Laboratory of Advanced Functional Porous Materials, Institute for New Energy Materials and Low-Carbon Technologies, Tianjin University of Technology, Tianjin, China. [6]Division of Materials and Manufacturing Science, Graduate School of Engineering, Osaka University, Suita, Osaka, Japan. [7]Mathematics for Advanced Materials Open Innovation Laboratory, AIST, Sendai, Japan. [8]Department of Materials Science and Engineering, Johns Hopkins University, Baltimore, MD, USA. ✉e-mail: hanjh08@gmail.com; mwchen@jhu.edu; hikato@imr.tohoku.ac.jp

intermetallic compounds with ordered atomic structures and well-defined stoichiometry possess the merits of homogeneous and intensified active sites as well as the capability to form unique crystal structures not commonly demonstrated by random solid-solutions to enhance the geometry effect[10,14,15]. The combination of electron localization and directional covalent bonding in intermetallic catalysts can also strengthen the electronic effect[14]. Moreover, the strong ionic/covalent interactions between the metal constituents are expected to enhance the chemical/electrochemical stability of intermetallic catalysts during catalytic processes. Intermetallic compounds, such as $Ni_4Mo$[16], $Co_3Mo$[10], $NiZn$[17], and $Ni_3Al$[18], are active HER electrocatalysts with appreciable energy efficiencies. Apart from the selection of intermetallic compounds with high intrinsic HER activity, a large specific surface area facilitating reactant adsorption, reaction, and desorption is essential for effectively utilizing the high catalytic activity of an intermetallic catalyst. In addition to the extensively studied particulate catalysts, bicontinuous porous nanostructures, with large surface areas, open pore channels, and high electrical conductivities, effectively maximize the density and accessibility of active sites and have emerged as a new type of electrocatalyst[19–24]. Three-dimensional (3D) bicontinuous nanoporous structures fabricated by dealloying[25], the selective removal of the less stable component(s) from an alloy, are very attractive owing to their large specific surface area with ample surface defects, high electric conductivity, and rapid mass transport pathways[19,26]. Several intermetallic compounds have been fabricated or integrated into 3D porous structures by the chemical etching of heterogeneous precursors (such as $Mg_2Cu$[27], $PtSi$[28], $(Pt_{1-x}M_x)_3Al$ (M = Ni, Co, Fe)[29–31], $SbSn$[32], and $Pd_3Bi$[24]) or by segregation as nanoparticulate phases anchored on 3D nanoporous metal frameworks during dealloying (such as $Cu_3Sn/Cu$[33], $Co_3Mo/Cu$[10], $Al_7Cu_4Ni/Cu$[34], and $Nb_5Si_3/NbTi$[35]). However, these porous intermetallic materials exhibit low flexibility in microstructural optimization, high electric resistance due to copious grain boundaries and interfaces, and poor mechanical stability. Although monolithic nanoporous intermetallic compounds with tunable bicontinuous nanoporosity are desirable for efficient electrocatalysis in energy-related reactions, fabricating bicontinuous nanoporous intermetallic catalysts from single-phase precursors remains a challenge because of the difficulty in forming chemically ordered intermetallic phases during low-temperature chemical dealloying. Accordingly, advanced HER electrocatalysts, based on non-noble nanoporous intermetallics with sufficiently high HER activity and robustness, are yet to be developed.

In this work, we report a liquid metal dealloying (LMD) strategy, conducted at elevated temperatures, to fabricate 3D nanoporous Mo-based intermetallic compounds as high-performance HER electrocatalysts. Coupled with the simultaneous evolution of 3D bicontinuous open porosity during the dealloying of single-phase precursors, the high-temperature LMD process overcomes the energy barrier, driving the direct formation of chemically ordered intermetallic phases. Beyond the conventional wisdom that porous metals produced by LMD typically have large feature sizes and therefore small specific surface areas for catalytic applications, here, the intermetallic $Co_7Mo_6$ and $Fe_7Mo_6$ exhibit fine nanoporous structures with ligaments/pores that are ~30 nm in size. Combined experimental analyses and molecular dynamics simulations uncovered an intermetallic effect, whereby high-temperature chemical ordering during nanopore formation suppresses dealloyed-nanostructure coarsening. The ultrafine 3D nanoporous $\mu$-$Co_7Mo_6$, with a large accessible surface area and abundant active sites, exhibits high HER catalysis in 1 M KOH alkaline electrolyte.

## Results and discussion
### Design, fabrication, and structural characterization
LMD is a processing technique utilizing the difference in the miscibility of alloy components in a molten metal bath to corrode selected component(s) while retaining the others, self-organizing into a 3D

porous structure[36–39]. According to the design principle (Fig. 1a), in the precursor alloy (AB), the pore-forming metal (A) and sacrificial component (B) should have a positive and negative enthalpy of mixing with the melt bath (C), respectively. LMD systems were designed accordingly to fabricate nanoporous Mo-based intermetallic compounds. A mixture of Mo with Co, Fe, or Cr, with designated compositions, was used as the pore-forming component A, while Ni and Mg were chosen as the sacrificial component B and melt bath C, respectively[36,37]. As shown in Fig. 1b, Mo, Fe, Cr, and Co have a positive enthalpy of mixing with Mg, whereas the enthalpy of mixing is negative between Ni and Mg, satisfying the principle of LMD[40]. The ternary alloy precursors for LMD were designed to have compositions of $Ni_{70}(Co_{0.55}Mo_{0.45})_{30}$, $Ni_{70}(Fe_{0.58}Mo_{0.42})_{30}$, and $Ni_{70}(Cr_{0.50}Mo_{0.50})_{30}$ (atomic ratios) (Supplementary Table 1), with the objective of fabricating nanoporous intermetallic $Co_7Mo_6$ and $Fe_7Mo_6$ and solid-solution $Cr_{50}Mo_{50}$ phases. Binary alloys of $Ni_{70}Mo_{30}$ and $Ni_{70}Fe_{30}$ were also prepared for fabricating nanoporous Mo and Fe, as references. All the precursors are single-phase face-centered cubic (fcc) alloys, as verified by X-ray diffraction (XRD) (Supplementary Fig. 1).

The $Ni_{70}(Co_{0.55}Mo_{0.45})_{30}$ system is used as an example to depict the fabrication of nanoporous intermetallic compounds via LMD. LMD was carried out by immersing a 200 μm-thick $Ni_{70}(Co_{0.55}Mo_{0.45})_{30}$ sheet sample into a 973 K Mg melt. During this process, Ni selectively dissolved into Mg, and Co-Mo spontaneously evolved into a bicontinuous porous solid structure at the alloy/melt interface. Owing to the fast dealloying velocity (in the order of 5–10 μm s$^{-1}$)[39,41], dealloying was completed in a short time (120 s). After cooling, the Mg melt was frozen in the porous structure, which was subsequently etched away using a dilute acid to expose the Co-Mo open porous structure. The changes of macroscopic morphology, phase, and microstructure are traced using optical photographs, XRD and scanning electron microscopy (SEM) images as shown in Fig. 1c–g, respectively. The fcc precursor $Ni_{70}(Co_{0.55}Mo_{0.45})_{30}$ exhibits an average grain size of ~15 μm and a uniform distribution of Ni, Co, and Mo (Fig. 1e and Supplementary Fig. 2). Significantly, the intermetallic compound $\mu$-$Co_7Mo_6$ (JCPDS 29-0489) together with the solidified Mg phase (JCPDS 35-0821) can be detected after LMD (Fig. 1d). The corresponding SEM image shows a bicontinuous nanocomposite consisting of interconnected $\mu$-$Co_7Mo_6$ (bright contrast) and Mg (dark contrast) phases, which are the characteristic of nanoporous structures generated by dealloying (Fig. 1f). The feature/ligament size is measured to be 30.8 nm, using the Fiji package of the ImageJ software (Supplementary Fig. 3)[42]. After Mg removal, nanoporous $\mu$-$Co_7Mo_6$ with a well-maintained bicontinuous structure is obtained (Fig. 1g). In agreement with the nanoscale ligament/pore size, the XRD pattern of nanoporous $\mu$-$Co_7Mo_6$ shows broad diffraction peaks (Fig. 1d). Notably, the final nanoporous $\mu$-$Co_7Mo_6$ inherits the free-standing sheet morphology of the precursor alloy with a well-preserved 2 cm × 1 cm lateral dimension (Fig. 1c and Supplementary Fig. 4). Due to LMD-induced volume contraction, the thickness of the sheet decreases from 200 μm to 170 μm. The nanoporous $\mu$-$Co_7Mo_6$ is further characterized by transmission electron microscopy (TEM). The bright-field TEM image shows interconnected ligaments composed of crystallites of ~30 nm in size (Fig. 1h). Intriguingly, this polycrystalline structure with nanosized grain ligaments is distinct from the typical microstructure of nanoporous gold dealloyed from Au-Ag alloys, which often retains the original grain structure of parent alloys[43,44]. Such distinction is likely owing to the difference in dealloyed phase formation between two alloy systems. The Au-Ag alloy system can form an ideal solid solution and has very small difference in atomic volumes between Au and Ag, allowing for an essentially coherent transformation from parent alloy to nanoporous product structure. In contrast, the LMD of nanoporous $\mu$-$Co_7Mo_6$ is fulfilled by the phase transformation from fcc-$Ni_{70}(Co_{0.55}Mo_{0.45})_{30}$ to trigonal $\mu$-$Co_7Mo_6$ with distinctly different crystal structures and lattice parameters (Supplementary Table 2). Consequently, the development of $\mu$-

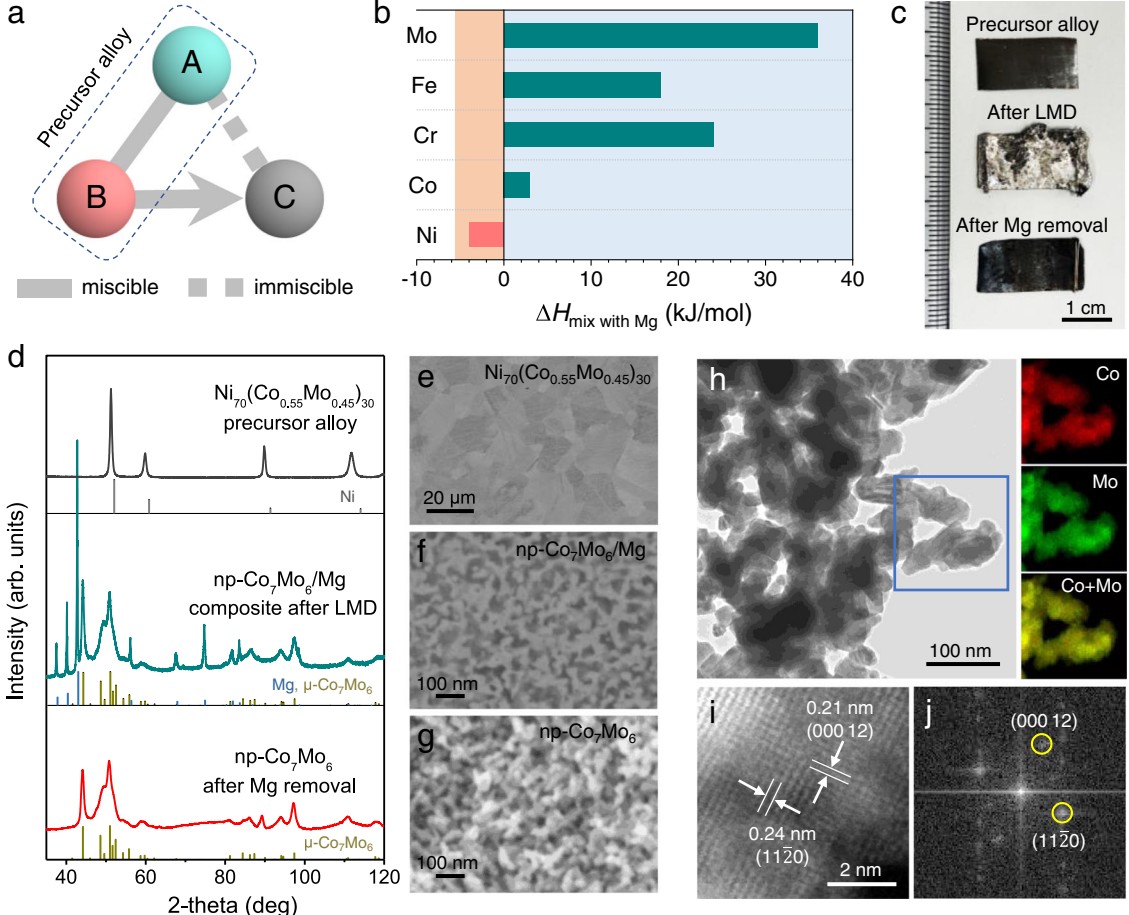

**Fig. 1 | Design and fabrication of the nanoporous intermetallic compounds.**
**a** Principle of LMD. **b** Enthalpy of mixing between the considered elements and the Mg melt. **c** Optical images, **d** XRD patterns, and **e**–**g** cross-sectional SEM images of the ternary precursor alloy $Ni_{70}(Co_{0.55}Mo_{0.45})_{30}$, np-$Co_7Mo_6$/Mg composite obtained after LMD, and np-$Co_7Mo_6$ obtained after further Mg removal. Standard patterns of Ni (PDF#65-0380), Mg (PDF#35-0821), and μ-$Co_7Mo_6$ (PDF#29-0489) are also shown in **d** as references. **h** Bright-field TEM image of np-$Co_7Mo_6$ and STEM-EDX elemental mapping for the selected area. **i** HAADF-STEM image and **j** corresponding FFT pattern of np-$Co_7Mo_6$.

$Co_7Mo_6$ ligaments during the dealloying of fcc-$Ni_{70}(Co_{0.55}Mo_{0.45})_{30}$ cannot be coherent and epitaxial, but is more likely through an interfacial heterogeneous nucleation and growth process, resulting in the formation of polycrystalline ligaments with nanosized grains. Similar polycrystalline structures had been observed in nanoporous gold produced by chemical dealloying of amorphous $Au_{40}Cu_{28}Ag_7Pd_5Si_{20}$ alloy[45], nanoporous Si by LMD of $Mg_2Si$ alloy[46], and nanoporous Ni by vapor phase dealloying of $Ni_2Zn_{11}$ alloy[47]. Despite being polycrystalline and inherently brittle, the nanoporous μ-$Co_7Mo_6$ maintains structural integrity and is mechanically robust due to the interconnections of ligaments in 3D space (Fig. 1c, Supplementary Figs. 4 and 5). The chemical composition of nanoporous μ-$Co_7Mo_6$ is verified by energy-dispersive X-ray spectroscopy (EDX) mappings, showing the uniform distribution of Co and Mo in the ligaments (Fig. 1h), which is in accordance with the observation of single-phase, as detected by XRD. Figure 1i and j show the atomic-resolution high-angle annular dark-field scanning transmission electron microscopy (HAADF-STEM) image of nanoporous μ-$Co_7Mo_6$ and the corresponding fast Fourier transform (FFT) pattern, respectively. The clear lattice fringes with $d$-spacings of 0.24 and 0.21 nm, highlighted in Fig. 1i, correspond to the ($11\bar{2}0$) and (000 12) crystal planes of the trigonal μ-$Co_7Mo_6$ (space group 166, R-3m)[48]. The X-ray photoelectron spectroscopy (XPS) spectra collected from the surfaces of nanoporous μ-$Co_7Mo_6$ show predominantly metallic Co and Mo signals along with some signals corresponding to oxidized $Mo^{4+}$, $Mo^{5+}$, $Mo^{6+}$, and $Co^{2+}$ species, most likely formed by air-

exposure (Supplementary Fig. 6). According to the Pourbaix diagrams of Mo and Co[49,50], these oxidized species can be reduced to metallic states during electrolysis on reduction polarization.

The microstructural characterizations confirmed the formation of monolithic 3D nanoporous structure for the intermetallic compound μ-$Co_7Mo_6$. This is the first report of the direct fabrication of a nanoporous intermetallic compound by LMD, via atomic self-assembly at the alloy/melt dealloying fronts. As a reference, it is found that the $Ni_{70}(Co_{0.55}Mo_{0.45})_{30}$ alloy cannot be dealloyed chemically in 0.5 M $H_2SO_4$, HCl, or $HNO_3$ solutions at room temperature (Supplementary Fig. 7), due to the low ambient-temperature surface-diffusivity of Mo and surface passivation after leaching the top few atomic layers[51]. Therefore, the LMD is of vital importance in fabricating nanoporous intermetallic compounds, owing to the elevated dealloying temperatures that accelerate atomic diffusivity for successful dealloying and drive the formation of chemical order for the intermetallic phases.

Under identical LMD conditions (973 K for 120 s), nanoporous $Fe_7Mo_6$, $Cr_{50}Mo_{50}$, Mo, and Fe were also fabricated from their ternary or binary alloy precursors (Fig. 2 and Supplementary Fig. 8). Nanoporous $Fe_7Mo_6$ is a μ-phase intermetallic compound (JCPDS 31-0641) with the same trigonal lattice structure as that of nanoporous μ-$Co_7Mo_6$[52]. Nanoporous $Cr_{50}Mo_{50}$ consists of two body-centered cubic (bcc) phases of Cr-Mo solid-solutions, with the compositions of $Cr_{25}Mo_{75}$ and $Cr_{88}Mo_{12}$ as estimated by the Vegard's law[53]. Both

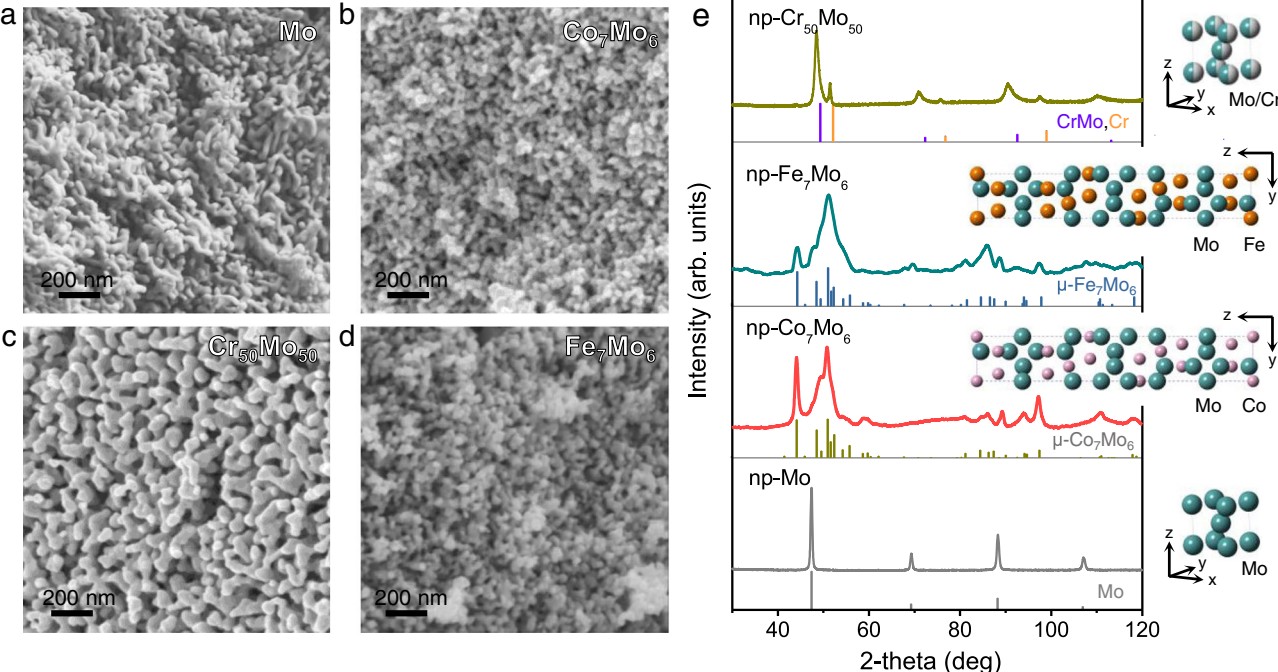

**Fig. 2 | Structural characterization of the nanoporous intermetallic compounds (or alloys).** SEM images of **a** np-Mo, **b** np-Co$_7$Mo$_6$, **c** np-Cr$_{50}$Mo$_{50}$ and **d** np-Fe$_7$Mo$_6$ prepared by LMD at 973 K for 120 s**. e** XRD patterns of the np-Mo, np-Co$_7$Mo$_6$, np-Cr$_{50}$Mo$_{50}$, and np-Fe$_7$Mo$_6$. Standard patterns of Mo (PDF#42-

1120), μ-Co$_7$Mo$_6$ (PDF#29-0489), μ-Fe$_7$Mo$_6$ (PDF#31-0641), Cr (PDF#06-0694), and Cr-Mo (PDF#65-9000) are also shown as references. Insets: Unit cell models of the corresponding phases: bcc-Mo, μ-Co$_7$Mo$_6$, μ-Fe$_7$Mo$_6$, and bcc-CrMo.

nanoporous Mo and Fe are bcc phases. The chemical compositions of these nanoporous materials, measured by SEM-EDX, are consistent with those of their corresponding phases (Supplementary Table 1). For all porous samples, the residual Ni and Mg contents remain below 6 at.%, a reasonable value for dealloyed materials. All five samples have uniform and well-defined 3D bicontinuous nanoporous structures (Fig. 2a–d and Supplementary Fig. 8). The average ligament/pore sizes are 30.8, 31.8, 50.3, 35.5, and 652.0 nm for nanoporous μ-Co$_7$Mo$_6$, μ-Fe$_7$Mo$_6$, Cr$_{50}$Mo$_{50}$, Mo, and Fe, respectively. The large pore size of porous Fe is consistent with the general fact that LMD-generated porous structures usually have feature sizes on the scale of hundreds of nanometer, and even microns, owing to severe coarsening during high-temperature processing[39]. In contrast, the feature sizes of nanoporous μ-Co$_7$Mo$_6$, μ-Fe$_7$Mo$_6$, Cr$_{50}$Mo$_{50}$, and Mo are small. While the small ligament sizes of nanoporous Mo and Cr$_{50}$Mo$_{50}$ could be explained by their high melting points (2896 K for Mo and ~2464 K for Cr$_{50}$Mo$_{50}$) and hence low atomic diffusivities, nanoporous μ-Co$_7$Mo$_6$ and μ-Fe$_7$Mo$_6$ are unique materials, as they have significantly low melting points (1892 and 1920 K for μ-Co$_7$Mo$_6$ and μ-Fe$_7$Mo$_6$, respectively) comparable to that of Fe (1811 K) but ligament sizes comparable to those of Mo and Cr$_{50}$Mo$_{50}$. The ligament sizes are temperature-dependent and can be controlled in the range of 20–43 nm, by varying the dealloying temperature from 873 to 1073 K, for nanoporous μ-Co$_7$Mo$_6$ (Supplementary Fig. 9).

**The intermetallic effect on LMD**
The thermal coarsening of dealloyed nanoporous materials is a surface diffusion-controlled process[54,55]. Sieradzki et al.[51] proposed an empirical correlation describing the scaling relation between the ligament size and inverse homologous temperature, $1/T_H = T_m/T_{dealloy}$, where $T_m$ is the melting point of the pore-forming components (metals and alloys), and $T_{dealloy}$ is the dealloying temperature, as shown in Fig. 3a. A survey of the literature data indicates a general trend that electrochemical dealloying (ECD) occupies the high $1/T_H$ region, with ligament sizes mostly below 100 nm, while high-temperature LMD corresponds

to the low $1/T_H$ region, with ligaments of hundreds of nanometer in size[39,56]. The scaling correlation between the ligament size and $1/T_H$ applies to both ECD and LMD. Plotting our LMD results at various temperatures in this diagram (Fig. 3a and Supplementary Figs. 10–14), a reasonable fit of the data to the conventional trend for nanoporous pure metals bcc-Mo, bcc-Fe, and solid-solution bcc-Cr$_{50}$Mo$_{50}$ is observed, even though their ligament sizes are relatively smaller than the literature data because of the different dealloying times (120 s for this study and 10-20 min in the literature). Significantly, a distinct deviation from the conventional scaling relation is observed for nanoporous intermetallic compounds of both μ-Co$_7$Mo$_6$ and μ-Fe$_7$Mo$_6$ (Fig. 3a). To eliminate the effect of the dealloying time, nanoporous μ-Co$_7$Mo$_6$ and μ-Fe$_7$Mo$_6$ were also fabricated with 15 min of dealloying time (Supplementary Figs. 10 and 11). As shown in Fig. 3a (as the open symbols), the ligament/pore sizes remain small and continue to deviate significantly from the typical scaling relationship. This indicates a unique and previously unexplored effect of intermetallic compounds in hindering the thermal coarsening of dealloyed nanoporous structures at high temperatures. Note that although the formation of surface oxides could also suppress surface diffusion[57], this effect could be excluded here as the LMD and pore formation are conducted in a highly reducing molten Mg environment.

In contrast to pure metals and solid-solution alloys, μ-Co$_7$Mo$_6$ and μ-Fe$_7$Mo$_6$ intermetallic compounds have chemically ordered crystal structures with covalent bonding, rather than pure metallic bonding[58,59]. The μ-Co$_7$Mo$_6$ and μ-Fe$_7$Mo$_6$ have topologically close-packed (TCP) structures, with alternating Zr$_4$Al$_3$ and MgCu$_2$-type layers[48,52] (Supplementary Fig. 15). Such an ordered lattice structure, with solely tetrahedral interstices[60], renders much slower atomic mobility than simple fcc or bcc metals. Additionally, covalent bonding with a high bond energy makes it difficult for atoms to hop[52,58,59], forming another barrier for atomic diffusion. The suppressed atomic diffusivity of the intermetallic compounds is confirmed by molecular dynamics simulations. The 14 nm × 15 nm × 8 nm models were built by sandwiching an 8 nm-thick slat of μ-Co$_7$Mo$_6$,

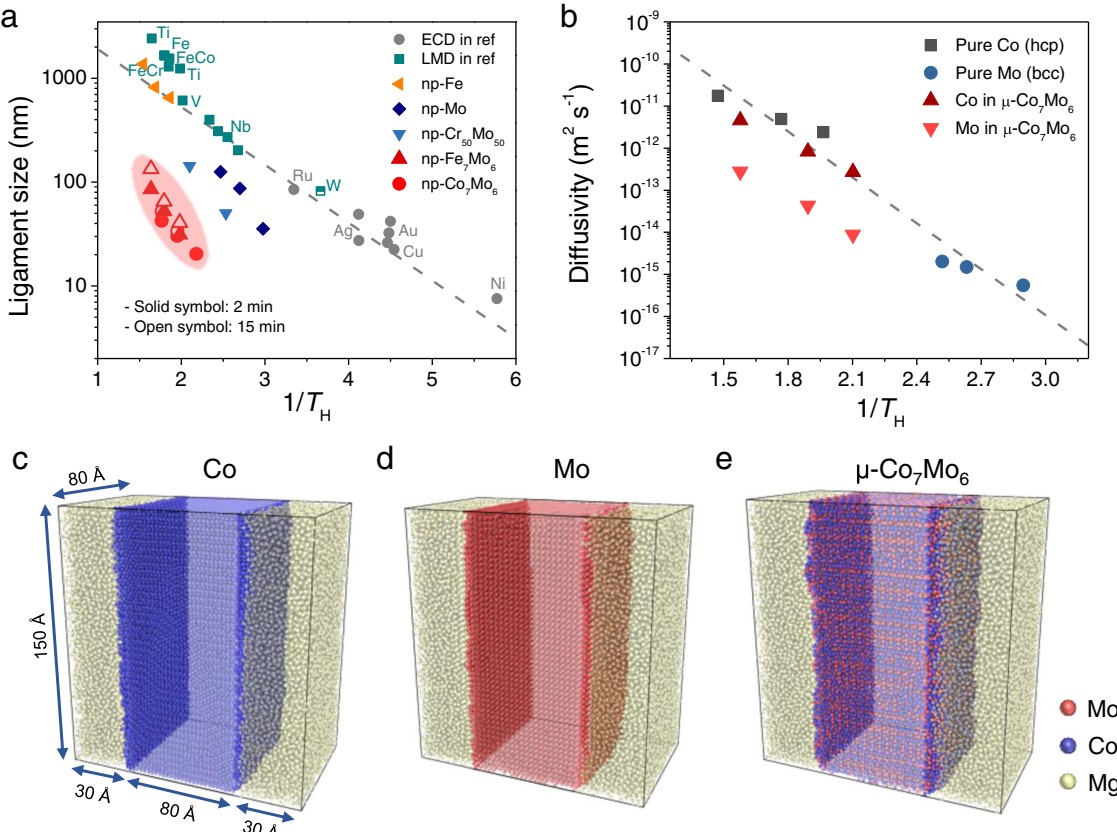

**Fig. 3 | Characteristic lengths of the nanoporous intermetallic compounds (or alloys) and molecular dynamics simulations of the interfacial diffusivities.**
**a** Ligament size versus the inverse homologous temperature, $1/T_H$ (i.e., $T_m/T_{dealloy}$), for the nanoporous intermetallic compounds (or alloys) and the elemental metals. Data for other porous metals fabricated by LMD (cyan squares) and ECD (gray circles) adapted from a reference[56] are included for comparison. Dealloying time is 10 min for LMD data from references, except for W, which is 20 min. For data from the present study, the dealloying time is 2 min for the solid symbols and 15 min for the open symbols. **b** Average interfacial diffusivities of pure Co (hcp), pure Mo (bcc), and Co and Mo in $\mu$-Co$_7$Mo$_6$ at 900–1200 K calculated by molecular dynamics simulations plotted against $1/T_H$. **c**–**e** Snapshots of the 3D models equilibrated at 1000 K, with molten Mg. Facets are (0001), (110), and (11$\bar{2}$0) for Co (hcp), Mo (bcc), and $\mu$-Co$_7$Mo$_6$, respectively, in these models. The three atomic layers at the metal/Mg melt interfaces are highlighted. The red, blue, and yellow balls represent the Mo, Co, and Mg atoms, respectively.

bcc-Mo, or hexagonal close packed (hcp)-Co, with designated facets ((11$\bar{2}$0), (1$\bar{1}$00), and (0001) for $\mu$-Co$_7$Mo$_6$; (100) and (111) for bcc-Mo; (10$\bar{1}$0) and (0001) for hcp-Co), between two 3 nm-thick molten Mg layers (Fig. 3c–e). The models were relaxed in the 900–1200 K temperature range, and the mean square displacement over a duration of 20 ns was calculated for the three atomic layers at the metal/Mg melt interfaces to derive interfacial diffusivity, according to the Einstein-Smoluchowski relation[61] (Supplementary Fig. 16). The obtained average diffusivities are plotted against $1/T_H$, as shown in Fig. 3b. The surface diffusivities of bcc-Mo, hcp-Co, and Co in $\mu$-Co$_7$Mo$_6$ follow a unified linear trend, but Mo in $\mu$-Co$_7$Mo$_6$ shows an apparent downward variation. Considering that the ligament size ($d(t)$) is related to surface diffusivity ($D_s$) according to $d(t)^n = KtD_s$ (where $K$ is a constant; $t$ is the dealloying time; and $n$, approximately 4, is the coarsening exponent)[47,62,63], the result is in good agreement with the trend of ligament size versus $1/T_H$, as shown in Fig. 3a. Therefore, the ultrafine pore/ligament sizes of the nanoporous intermetallic compounds originate from suppressed Mo atomic diffusivity, particularly at the surfaces and interfaces of the intermetallic compounds. Based on the calculated diffusivities, activation energies of surface diffusion are determined to be 59.8, 84.2, 84.7, and 102.5 kJ mol$^{-1}$ for hcp-Co, bcc-Mo, Co in $\mu$-Co$_7$Mo$_6$, and Mo in $\mu$-Co$_7$Mo$_6$, respectively (Supplementary Table 3). Therefore, the intermetallic phase provides a large energy barrier for the surface diffusion of Co and Mo atoms.

## Mechanisms for the evolution of ultrafine nanoporous intermetallic compounds

Based on the above experimental analyses and molecular dynamics simulations, a scheme for fabricating ultrafine nanoporous intermetallic compounds by LMD is outlined in Fig. 4. Taking nanoporous $\mu$-Co$_7$Mo$_6$ as an example, immersing the single-phase Ni$_{70}$(Co$_{0.55}$Mo$_{0.45}$)$_{30}$ alloy into the Mg melt selectively dissolves the Ni atoms, leaving behind Co and Mo atoms at the alloy/melt interfaces, which diffuse and self-assemble into a 3D nanoporous solid structure. Despite the thermodynamic advantages, the formation of an ordered intermetallic phase is kinetically hampered by a high energy barrier[14]. The LMD overcomes this energy barrier by virtue of its high temperatures, enabling the concurrent formation of the chemically ordered $\mu$-Co$_7$Mo$_6$ phase and bicontinuous nanoporosity, which facilitate the direct formation of nanoporous intermetallic compounds. In the dealloying process, nanopore formation is followed by thermal coarsening, which is a surface-diffusion-controlled process driven by thermodynamic surface-energy reduction[39,64,65]. For conventional metals and random solid-solution alloys with moderate energy barriers of surface diffusion, high-temperature LMD drives severe thermal coarsening because of the significantly accelerated surface diffusivity, forming very coarse porous structures with pore/ligament sizes larger than hundreds of nanometer. In contrast, intermetallic compounds have a much larger energy barrier for surface diffusion, which regulates slow surface diffusivity even

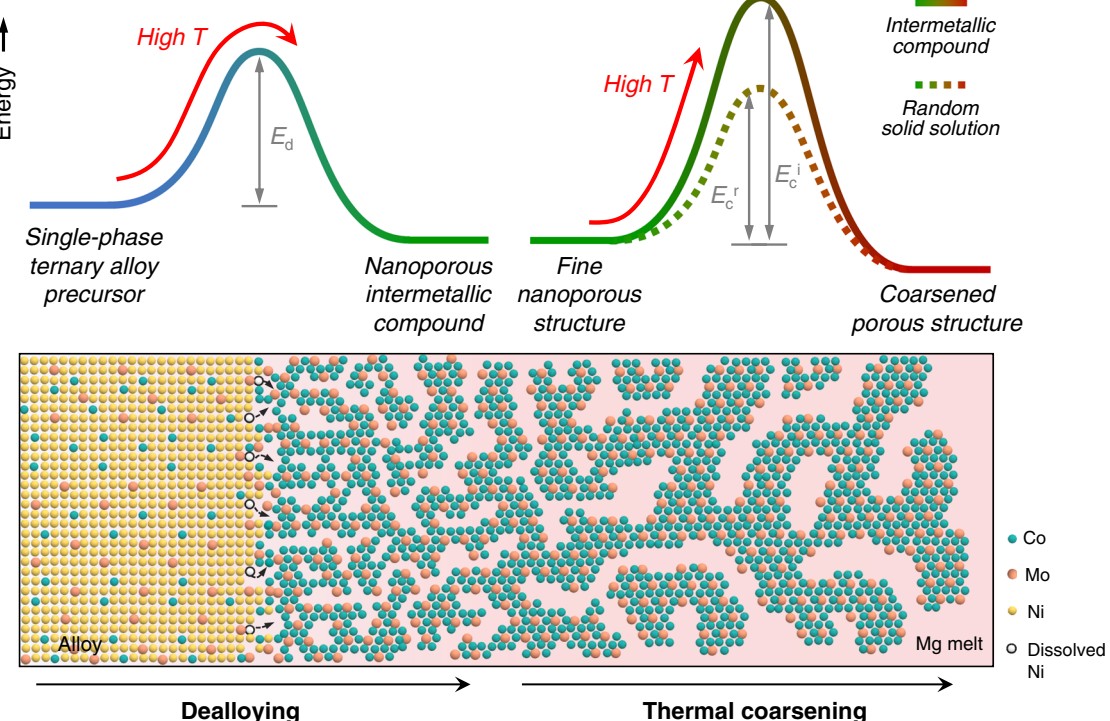

**Fig. 4 | Schematic illustrations of the fabrication of ultrafine nanoporous μ-Co₇Mo₆ intermetallic compound by LMD.** The process involves dealloying, to create a nanoporous structure of an ordered intermetallic phase, and subsequent thermal coarsening, to increase the feature size of the porous structure. $E_d$ represents the energy barrier of dealloying to form the nanoporous intermetallic compound. $E_c^r$, and $E_c^i$ represent energy barriers of thermal coarsening for random solid-solution alloys and intermetallic compounds, respectively. The green, orange, yellow and white balls represent the Co, Mo, Ni, and dissolved Ni atoms, respectively.

under high-temperature LMD conditions, significantly suppressing dealloyed-nanostructure coarsening and forming nanoporous intermetallic compounds with an ultrafine feature size. In the typical low $1/T_H$ region of LMD, nanoporous μ-Co₇Mo₆ and μ-Fe₇Mo₆ show very small ligament sizes of 20–100 nm, which are up to two orders of magnitude smaller than those of conventional nanoporous metals processed by LMD (Fig. 3a). Therefore, the intermetallic effect enables the utilization of high-temperature LMD for ordered-intermetallic-phase formation, while maintaining a fine nanoporous structure by circumventing the severe thermal coarsening. This is particularly advantageous for fabricating 3D nanoporous intermetallic compounds for electrocatalytic reactions, as fine structures with large surface areas and abundant under-coordinated surface defects contribute to high catalytic activity[66,67].

**Electrochemical characterizations**

The electrocatalytic performance of the nanoporous intermetallic compounds for the HER was evaluated in a 1 M KOH aqueous solution at room temperature. Two forms of the intermetallic catalysts were investigated: powder samples and self-supported monolithic sheets. The powder samples were prepared from as-dealloyed nanoporous sheets by mechanical grinding and high-power ultrasonication (Supplementary Fig. 17). As a mixture with conductive carbon and nafion binder, they were cast onto a glassy carbon disk electrode to make thin films with a 2 mg cm⁻² catalyst loading. Such thin film electrodes can reduce the ion diffusion constraints that thick nanoporous sheet electrodes may suffer, allowing for a more precise assessment of the intrinsic activity of nanoporous electrocatalysts. Figure 5a–d illustrate the electrochemical HER performance of nanoporous Mo-based intermetallic compounds and the reference materials. All nanoporous catalysts were fabricated at the temperature of 973 K. Based on the multiple criteria of onset potential (Fig. 5a), Tafel slope (Fig. 5b, c), operating

potential at 10 mA cm⁻²$_{geo}$ current density (Fig. 5a, c), and charge transfer resistance ($R_{ct}$) from electrochemical impedance spectroscopy (EIS) measurements (Fig. 5d), the catalytic activities exhibit the following trend: Pt/C > μ-Co₇Mo₆ > μ-Fe₇Mo₆ > Mo > Cr₅₀Mo₅₀. Pure carbon is essentially inert in the tested potential range. Among the nanoporous electrocatalysts, μ-Co₇Mo₆ demonstrats the lowest overpotential of 76 mV at 10 mA cm⁻²$_{geo}$ current density, the lowest Tafel slope of 82 mV dec⁻¹, and the smallest $R_{ct}$ of 5.1 Ω, indicating the highest HER kinetics and catalytic efficiency. The catalytic activity of nanoporous μ-Co₇Mo₆ is close, though yet inferior, to that of Pt/C.

To assess the intrinsic activity of the catalysts, the electrochemical surface areas (ECSAs) of the nanoporous electrocatalysts were calculated by measuring their double layer capacitances ($C_{dl}$) (Supplementary Fig. 18). The ECSAs for nanoporous μ-Co₇Mo₆, μ-Fe₇Mo₆, Cr₅₀Mo₅₀, and Mo are determined to be 44.6, 54.9, 9.3, and 16.9 m² g⁻¹, respectively, clearly demonstrating an abundance of active sites in the fine-structured nanoporous intermetallic compounds. The ECSA-normalized HER polarization curves, in Supplementary Fig. 18h, indicate an intrinsic catalytic activity trend of the order of μ-Co₇Mo₆ > Mo > Cr₅₀Mo₅₀ > μ-Fe₇Mo₆. The activity trend for μ-Co₇Mo₆, Mo, and Cr₅₀Mo₅₀ remains unchanged compared to the geometric-surface-area-normalized results. However, nanoporous μ-Fe₇Mo₆ exhibits lower intrinsic activity but higher apparent activity (on a geometric surface area basis) than nanoporous Mo and Cr₅₀Mo₅₀, indicating that the large surface area and associated abundant active sites originating from a small ligament size of nanoporous μ-Fe₇Mo₆ contribute to the enhanced catalytic performance. In contrast to μ-Fe₇Mo₆, the nanoporous μ-Co₇Mo₆ has both the highest intrinsic activity and a fine porous structure for a large surface area, resulting in the highest catalytic activity among the nanoporous electrocatalysts. The superior HER activity of μ-Co₇Mo₆ over μ-Fe₇Mo₆ could be attributed to the electronic effect that the stronger *d*-electron

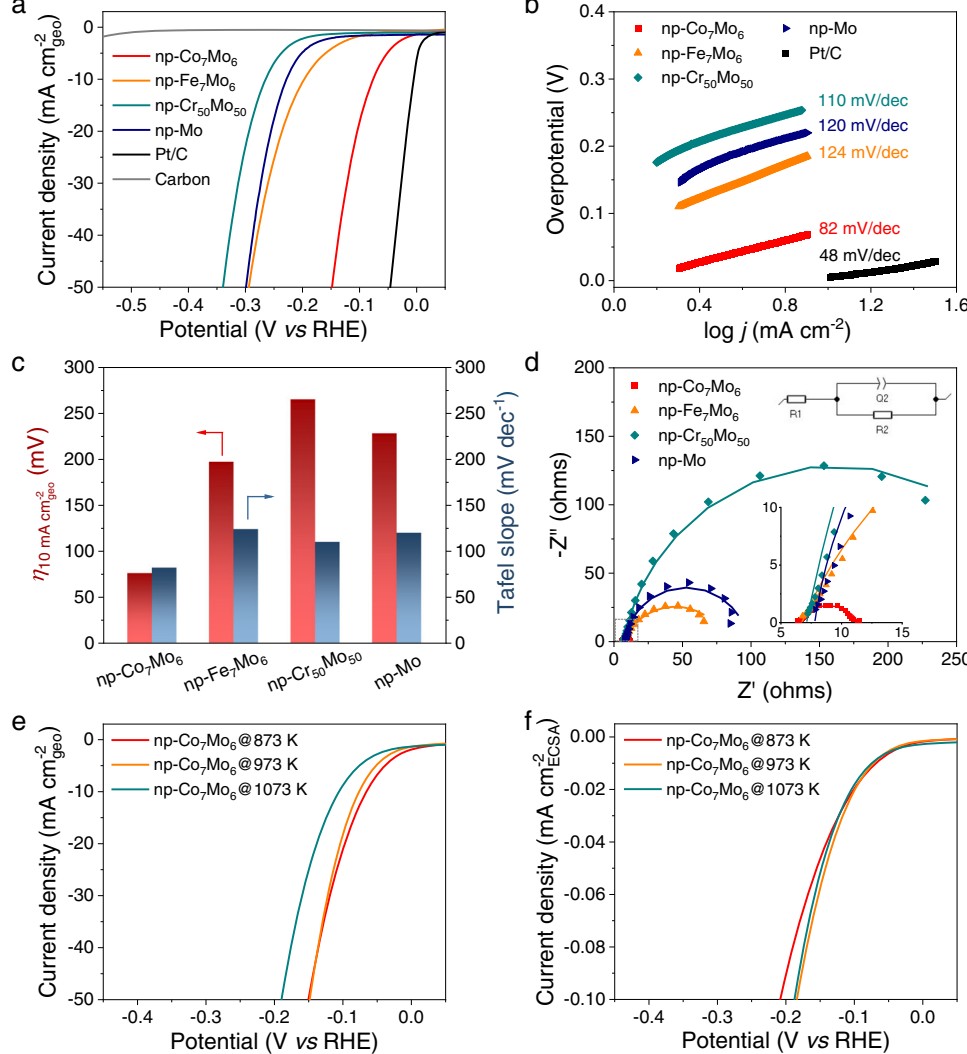

**Fig. 5 | Electrochemical HER performance of the nanoporous Mo-based intermetallic compounds and the reference materials tested as powder samples cast on a disk electrode. a** iR-corrected HER polarization curves. **b** Tafel plots. **c** Comparisons of overpotential at 10 mA cm$^{-2}_{geo}$ current density and Tafel slopes. **d** EIS spectra collected at 200 mV overpotential. The insets are the zoom-in spectra and the electronic circuit for EIS fitting. **e, f** iR-corrected HER polarization curves with current normalized by the geometric surface area and the electrochemical active surface area (ECSA), respectively, for np-Co$_7$Mo$_6$ fabricated by LMD at the temperatures of 873 K, 973 K, and 1073 K. The catalyst loading is 2 mg cm$^{-2}$ for all samples except for pure carbon.

interactions between Co and Mo in this hypo−hyper-*d*-electronic combinations modulate the optimal adsorption/desorption of reaction intermediates[68,69].

The nanoporous µ-Co$_7$Mo$_6$ fabricated by LMD at various temperatures were also examined as powder samples to determine the effect of pore/ligament size on the electrochemical properties. The three samples prepared at 873 K, 973 K, and 1073 K exhibit ligament diameters of 20.7, 30.8, and 43.1 nm, respectively (Supplementary Fig. 9 and Supplementary Table 4). As depicted in Fig. 5e and Supplementary Fig. 19, lower dealloying temperatures and smaller ligament sizes result in larger HER current densities on a geometric surface area basis for a given overpotential, indicating higher catalytic activity. The measured ECSAs for these electrocatalysts are 53.5, 44.6, and 24.1 m$^2$ g$^{-1}$, respectively (Supplementary Fig. 20). Remarkably, the ECSA-normalized HER polarization curves of the three electrocatalysts overlap when the current density is less than 0.037 mA cm$^{-2}_{ECSA}$ (equivalent to 40 mA cm$^{-1}_{geo}$ for the 873 K sample), suggesting that they all possess the same intrinsic catalytic activity (Fig. 5f). Therefore, the smaller-ligament-induced increment in apparent catalytic efficiency is due to a structural effect in which a larger specific surface area provides a larger number of catalytically active sites for

electrochemical reactions. This structural effect highlights the significance of the intermetallic effect in realizing the nanoscale porosity structure of Mo-based intermetallic compounds for superior electrochemical performance. Note that when the current density exceeds 40 mA cm$^{-1}_{geo}$, the 873 K sample's activity decreases and becomes lower than that of the 973 K sample (Fig. 5e, f). This is probable because the ultrafine porous structure (20.7 nm pore size) of the 873 K sample limits ion/gas transport at high current densities[70]. Under the current testing conditions, the 973 K sample with a pore size of 30.8 nm appears to be the optimal electrocatalyst capable of balancing the intrinsic reaction kinetics and accessibility of active sites.

The monolithic nanoporous intermetallic compounds are mechanically robust and can therefore be used as self-supported HER electrodes. Figure 6a shows the HER polarization curve of nanoporous µ-Co$_7$Mo$_6$ sheet electrodes with a dimension of 1 cm × 0.5 cm × 170 µm (Supplementary Fig. 21), in comparison to a Pt/C electrode drop-cast on carbon paper with a similarly high catalyst loading of 24.4 mg cm$^{-2}$. Due to intensified active sites from the bulk electrode, the nanoporous µ-Co$_7$Mo$_6$ sheet delivers improved HER performance compared to its powder counterpart, exhibiting a ~0 V onset potential, a 46 mV dec$^{-1}$ Tafel slope, and a 14 mV overpotential at 10 mA cm$^{-2}_{geo}$ current density

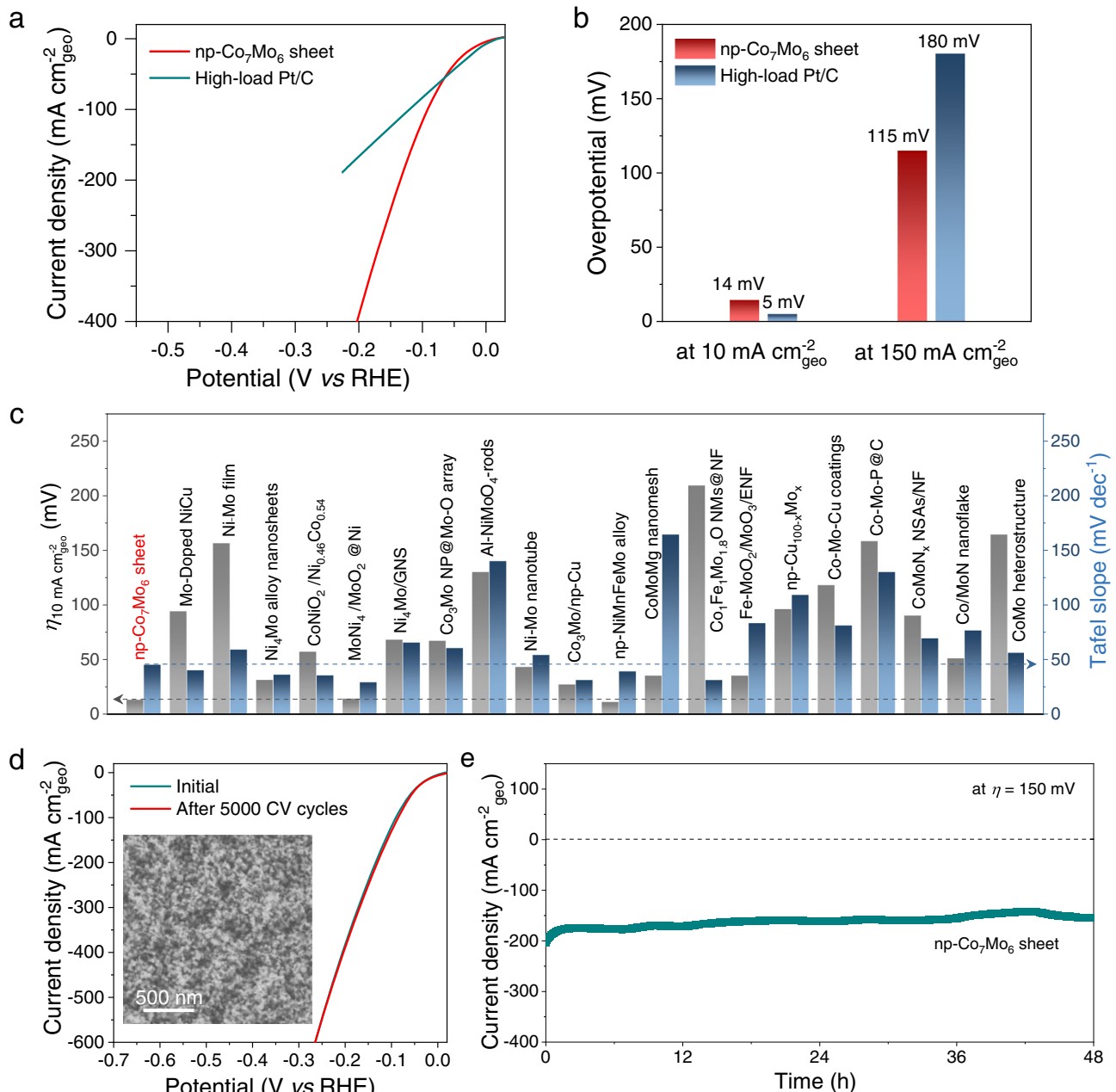

**Fig. 6 | HER catalytic performance and durability of the self-supported np-Co₇Mo₆ sheet electrocatalysts (fabricated at 973 K). a** iR-corrected HER polarization curve as compared to a high-load Pt/C catalyst. **b** Comparisons of overpotential at 10 and 150 mA cm⁻²geo current density. **c** Overpotential at 10 mA cm⁻²geo and Tafel slope of the np-Co₇Mo₆ sheet in comparison with other Mo-based HER electrocatalysts reported in the literature (also summarized in Supplementary Table 5). **d** iR-corrected HER polarization curves collected before and after the accelerated degradation test for 5000 cycles. Inset: SEM image of the np-Co₇Mo₆ sheet after the stability measurement. **e** Long-term stability measurements for 48 h at 150 mV overpotential.

(Fig. 6a, b and Supplementary Fig. 22). The property is among the best performances reported for non-precious Mo-based HER electrocatalysts (Fig. 6c and Supplementary Table 5). Due to the inferior intrinsic activity of nanoporous μ-Co₇Mo₆ compared to that of Pt/C, the μ-Co₇Mo₆ sheet exhibits larger overpotentials at current densities below 55 mA cm⁻²geo than Pt/C (Fig. 6a). In contrast, μ-Co₇Mo₆ outperforms Pt/C at high current densities exceeding 55 mA cm⁻²geo. For instance, an overpotential of 115 mV is required to obtain a current density of 150 mA cm⁻²geo for nanoporous μ-Co₇Mo₆, which is significantly lower than the 180 mV necessary for Pt/C (Fig. 6b). While the low-current-density property of the electrocatalysts is mainly controlled by kinetics, at higher current densities the density and accessibility of active sites can dominate the performance, particularly in the

porous electrodes. Therefore, the comparison between nanoporous μ-Co₇Mo₆ and Pt/C demonstrates that the 3D bicontinuous porous structure facilitates the electrochemical reactions of the intermetallic compounds by readily exposing more active sites, rendering the nanoporous μ-Co₇Mo₆ particularly suitable for high-current applications. Moreover, unlike Pt/C electrodes, which are prone to exfoliation due to the severe production of H₂ bubbles at high current densities, the robust self-supported μ-Co₇Mo₆ sheet with 3D interconnected ligaments has a far better structural stability. Along with excellent electrocatalytic activity, the nanoporous μ-Co₇Mo₆ electrodes exhibit excellent durability. After 5000 cycles of accelerated degradation testing by cyclic voltammetry, the nanoporous electrode shows a negligible shift in the HER polarization curves and negligible change in

its porous structure (Fig. 6d). Furthermore, the nanoporous μ-Co$_7$Mo$_6$ delivers a high current density of -170 mA cm$^{-2}$ at 150 mV over-potential, without significant current decay over a long-term chron-oamperometry test for 48 h (Fig. 6e). It is necessary to note that despite the enhanced catalytic efficiency of self-supported nanopor-ous μ-Co$_7$Mo$_6$, the sheet electrodes have a much lower material utili-zation efficiency than powder samples (Supplementary Fig. 23) because only the near surface layers of the thick electrodes may con-tribute to electrochemical reactions, particularly at a high current density. High material efficiency would necessitate further optimiza-tion of the sheet thickness and the nanoporous structure. In addition, advanced catalyst integration techniques must be developed in order to implement the intrinsically brittle nanoporous μ-Co$_7$Mo$_6$ sheet as large-sized electrodes in commercial electrolyzers. By contrast, the powder form of the nanoporous intermetallic electrocatalysts, which can be spray-coated on substrates to make scalable electrodes with high material efficiency[71], might be readily applicable for practical hydrogen production applications.

In summary, we presented a high-temperature LMD strategy for fabricating 3D nanoporous intermetallic catalysts of μ-Co$_7$Mo$_6$ and μ-Fe$_7$Mo$_6$. The formation of pore-forming intermetallic compounds leads to slow atomic diffusivity and retards thermal coarsening of nanopor-ous structures at high temperatures. The characteristic ligament sizes of nanoporous μ-Co$_7$Mo$_6$ and μ-Fe$_7$Mo$_6$ are only tens of nanometer, notably departing from the conventional scaling relationship between the ligament size and dealloying temperature. The nanoporous μ-Co$_7$Mo$_6$ with ultrafine porous structure exhibits high electrocatalytic activity and durability, indicating its potential use as a commercial HER catalyst for high-current applications. This study sheds light on the intermetallic effect on tailoring nanostructures of dealloyed materials and could assist the development of advanced nanoporous inter-metallic electrocatalysts for a broad range of energy applications.

## Methods

### Fabrication of nanoporous Mo-M (M = Co, Fe, Cr)

The master alloys with nominal compositions of Ni$_{70}$(Co$_{0.55}$Mo$_{0.45}$)$_{30}$, Ni$_{70}$(Fe$_{0.58}$Mo$_{0.42}$)$_{30}$, Ni$_{70}$(Cr$_{0.50}$Mo$_{0.50}$)$_{30}$, Ni$_{70}$Mo$_{30}$, and Ni$_{70}$Fe$_{30}$ (atomic ratios) were prepared from pure metals (Ni (99.9%) from FUJIFILM Wako Pure Chemical Co., Osaka, Japan; Co (99.99%) from Rare Metallic Co., Ltd., Tokyo, Japan; Fe (>99.95%) from Toho Zinc Co., Ltd., Tokyo, Japan; Cr (99.9%) from Kojundo Chemical Laboratory Co., Ltd., Saitama, Japan; Mo (99.95%) from Furuuchi Chemical Co., Tokyo, Japan) by arc melting in the argon atmosphere. The as-prepared ingots were re-melted in a tilt casting machine with argon protection and casted into thick plates with a thickness of 5 mm. The plates were further cold rolled into a thickness of 200 μm and then thermally treated at 1273 K for 8 h for homogenization and relieving the internal stress. The liquid metal dealloying (LMD) was carried out in a helium atmosphere using a magnesium melt (Mg (99.9%) from Rare Metallic Co., Ltd., Tokyo, Japan). The magnesium was melted and heated to temperatures of 973 K, 1073 K, or 1173 K by induction heating. The precursor alloy sheets of Ni$_{70}$(Co$_{0.55}$Mo$_{0.45}$)$_{30}$, Ni$_{70}$(Fe$_{0.58}$Mo$_{0.42}$)$_{30}$, Ni$_{70}$(Cr$_{0.50}$Mo$_{0.50}$)$_{30}$, Ni$_{70}$Mo$_{30}$, and Ni$_{70}$Fe$_{30}$ were immersed into the magnesium melt for a duration of 120 s. For LMD at 873 K, the eutectic Mg$_{90}$Ca$_{10}$ alloy melt was used in place of the Mg melt and a longer immersion time of 1800 s was used to ensure the complete dealloying of the entire sheet. After LMD, the obtained samples were etched in 0.05 M HCl in ethanol solution for 24 h to remove the solidified magnesium.

### Materials characterization

The X-ray diffraction was carried out using a Bruker D8 Discover diffractometer with Co-Kα radiation (Yokohama, Japan, λ = 1.7902 Å) at a scan rate of 0.5° per min. The microstructure and chemical composition were explored by a field-emission scanning electron microscope (Ultra 55, ZEISS, Oberkochen, Germany) coupled with an energy dispersive X-ray spectrometer (Quantax, Bruker, Billerica, MA, USA). The sample after LMD for cross-section imaging was pre-pared by mechanical polishing and ion milling (E-3500, Hitachi, Tokyo, Japan). Further investigations of the microstructure and chemical compositions were performed using a field-emission transmission electron microscope (JEOL JEM-2100F, 200 keV) with double spherical aberration correctors for both the probe-forming and image-forming objective lenses. Surface compositions of the samples were characterized by an X-ray photoelectron spectroscopy (XPS, AXIS ultra DLD, Shimazu) in a vacuum pressure of 10$^{-7}$ Pa with an Al Kα (mono) anode. The specific surface area was measured by N$_2$ adsorption–desorption method using BELSORP-mini II (BEL. JAPAN. INC) at 77.0 K.

### Electrochemical characterization

All the electrochemical measurements were performed in 1.0 M KOH aqueous solution at 293 K using a Bio-Logic VSP 300 electrochemical workstation and a three-electrode electrochemical cell with a Pt wire or graphite rod counter electrode and an Ag/AgCl reference electrode. The powder samples were prepared from as-dealloyed sheet samples using a combination of mechanical grinding with a mortar and pestle and high-power ultrasonication at 130 W. 10 mg of nanoporous sample powder, 2 mg of conductive acetylene black (MTI Corp.), and 50 μL of Nafion solution (Wako Corp., 5 wt%) were dispersed in 1 mL of ethanol and sonicated for one hour to obtain a homogeneous ink. 39.2 μL of the ink was then cast onto a glassy carbon disk electrode (5 mm in dia-meter). The benchmark Pt/C catalyst was prepared from commercial Pt/ C (20 wt%) using the same mixture composition. The catalyst loading of the nanoporous catalysts and Pt/C was 2 mg cm$^{-2}$. A pure carbon elec-trode with a carbon loading of 0.4 mg cm$^{-2}$ (equivalent to the loading of carbon components in the mixture electrodes) was also made by the same procedure without adding the nanoporous catalysts. For elec-trochemical measurements employing self-supported sheet samples, the sheet electrodes had a size of 1 cm × 0.5 cm × 170 μm. The high-load Pt/C catalyst was prepared by casting commercial Pt/C powders on a carbon paper. A total of 610 μL of the ink made by mixing 10 mg of Pt/C (20 wt%) and 125 μL of Nafion solution (5 wt%) in 1 mL of ethanol was drop-cast onto the carbon paper at a loading amount of 24.4 mg cm$^{-2}$ and dried at room temperature. The polarization curves were char-acterized by cyclic voltammetry (CV) with a scan rate of 2 mV s$^{-1}$. iR-compensation was applied to all polarization curves based on resistance data measured by EIS at −0.9 V vs Ag/AgCl. The potential versus Ag/AgCl was changed into RHE based on E$_{vs\,RHE}$ = E$_{vs\,Ag/AgCl}$ + E$_{Ø\,Ag/AgCl}$ + 0.059 pH. The electrochemical impedance spectroscopy (EIS) was carried out at −0.2 V vs RHE with the frequency from 10$^5$ Hz to 0.01 Hz. The accel-erated cyclic voltammetry cycling test was conducted at the scan rate of 50 mV s$^{-1}$ for 5000 cycles. The chronoamperometric responses were tested by holding the potential at −0.15 V vs RHE. The double-layer capacitance (C$_{dl}$) was measured by CV in non-Faradaic potential region (−0.9 V to −1.0 V vs Ag/AgCl) with scan rates of 10, 20, 30, 40 and 50 mV s$^{-1}$. The electrochemical surface area (ECSA) is calculated by subtracting the capacitive contributions from carbon:

$$ECSA = \left( C_{dl} - C_{dl@carbon} \right)/C_s \qquad (1)$$

where C$_{dl}$ is equivalent to the linear slope in the function of charging current density differences (Δj = |j$_a$-j$_c$ |/2) and scan rate and C$_s$ is the specific capacitance with a moderate value of 0.040 mF cm$^{-2}$$_{ECSA}$[10,72].

### Molecular dynamics simulation

Large-scale Atomic/Molecular Massively Parallel Simulator (LAMMPS)[73] was used to simulate the process of atom diffusion at the interfaces between Co (hcp), Mo (bcc), or Co$_7$Mo$_6$ (intermetallic compound, μ-phase) and a magnesium. A 14 nm × 15 nm × 8 nm model

was prepared by sandwiching an 8 nm-thick slat of μ-$Co_7Mo_6$, bcc-Mo or hcp-Co with designated facets (($11\bar{2}0$), ($1\bar{1}00$), and ($0001$) for μ-$Co_7Mo_6$; ($100$) and ($111$) for bcc-Mo; ($10\bar{1}0$) and ($0001$) for hcp-Co) between two 3 nm-thick molten Mg layers. The embedded atomic method (EAM) potential fitted by Zhou et al.[74] was used to represent interactions between atoms. The crystal/liquid models were relaxed at 1000 K for 20 ns and subsequently equilibrated at 900–1200 K for 200 ps with zero external pressure (NPT). Then, the systems were held at 900–1200 K for 20 ns under the constant volume condition (NVT). The three atomic layers at the interfaces were extracted and diffusivities of these atoms were calculated based on the mean square displacement (MSD) and the Einstein-Smoluchowski relation.

## Data availability

The raw/processed data generated in this study are available at Zenodo.org, https://doi.org/10.5281/zenodo.6982712. Source data are provided with this paper.

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

## Acknowledgements

This work was supported by the Collaborative Research Center on Energy Materials in the Institute of Materials Research (E-IMR), Tohoku University, and Grant-in-Aid for Scientific Research from the Japan Society for the Promotion of Science (JSPS) KAKENHI (Grant Nos. 21J12719, 20K05126, 19K15389, and 18H05939). M. Chen was supported by the Whiting School of Engineering, Johns Hopkins University, and the National Science Foundation (NSF DMR-1804320). R. Song acknowledges support from the China Scholarship Council. We thank Ms. Omura of the IMR at Tohoku University for the XPS tests.

## Author contributions

H.K., J.H. and M.C. conceived and supervised this research. R.S. and J.H. conducted materials preparation, characterization, and electrochemical measurements. M.O performed the MD simulations. A.K. performed the mechanical property measurements. R.S., J.H., M.O., R.B., T.W., J.J., D.W., Y.T., M.C. and H.K. analyzed the data. J.H. and R.S wrote the paper. All the authors discussed the results and reviewed the manuscript.

## Competing interests

The authors declare no competing interests.
