## [Peer Review File · Nature Communications]

Ultrafine Nanoporous Intermetallic Catalysts by
High-Temperature Liquid Metal Dealloying for Electrochemical
Hydrogen ProductionREVIEWER COMMENTS

Reviewer #1 (Remarks to the Author):

Presented in this manuscript is the development and application of intermetallic nanoporous alloys for hydrogen evolution. The authors present a unique approach to forming intermetallic alloy nanostructures which are difficult to form through other traditional methods. The demonstration of the mechanism of formation and pore structure stability at elevated temperatures will be a significant addition to the nanoporous metal and dealloying fields. The HER activity of the intermetallic alloys is competitive with a commercial Pt-based catalyst. Find detailed comments and questions below:

1. In the introduction it is mentioned that nanoporous bicontinuous materials have been used extensively for catalysis, however, there are no citations listed. There is a huge body of literature for nanoporous metals for catalysis and electrocatalysis, the authors need to add some of those citations here.
2. In the introduction and results/discussion, the authors need to add more detailed discussion of why intermetallic alloys are expected to be more catalytically active than random/homogeneous alloys. This would add more justification and support for their work.
3. It is noted that there is a significant difference in formed nanoporous metal surface area depending upon the precursor alloy composition and dealloying parameters. Also, it appears that carbon particles are added to the electrodes, likely to improve conductivity. Are the current densities and activities reported in Figures 5 and 6 normalized by the real active area of the material, or just the geometric surface area of the electrode? Since there are differences in material composition and active area, the activity should be normalized by the real active area as this will indicate whether these differences in activity are just related to the structure of the material or if there is a real kinetic impact of the nanoporous materials. Also, if the active area is determined by double layer capacitance, do the authors anticipate any error in this determinate due to incorporation of carbon particles into the electrodes?
4. The comparison of HER activity at high current density needs a bit more discussion. Low current density performance is controlled by kinetics, but at higher current densities, especially in porous electrodes, ion and gas transport as well as shifts in interfacial pH can dominate the performance. In this regard, can the authors comment on the different in performance of their material and Pt/C at higher current densities?
5. In Figure 1g and 1h, it appears that the nanoporous material is polycrystalline with nanosized grains. In traditional dealloying, the dealloyed material retains the crystal and grain structure of the parent alloy. Is it different here? It would add to the manuscript if the authors can add some discussion here.
6. The CoMo material is dealloyed at different temperatures to evolve different nanostructures. Is the composition the same for all nanoporous metals, regardless of temperature used to dealloy?

Reviewer #2 (Remarks to the Author):

The authors report on the fabrication of monolithic nanoporous intermetallic compounds via liquid metal dealloying (LMD) and their performance for electrocatalytic hydrogen production. Using starting alloys that result in the formation of intermetallic compounds during LMD is a clever idea, and experiments are well designed and beautifully executed. Using experiments and theory, the authors convincingly demonstrate that the ultra-fine feature size observed for intermetallic μ -Co₇Mo₆ and μ -Fe₇Mo₆ is a consequence of the suppressed Mo atomic diffusivity. The reviewer believes that these experiments alone justify publication in Nature Communications. However, the comments listed below should be considered before acceptance of the manuscript:

- 1) I could not find the information about the size of the samples they produced and used for the electrochemical experiments (only the thickness was provided)

2) No information about the mechanical properties is provided. For the ECSA experiments the authors used powders prepared by ball milling suggesting that the material is brittle. It is not necessary to add mechanical tests, but at least a qualitative description of the mechanical behavior and handling properties should be added.

3) My main critic is the apparent focus on high-performance electrochemical hydrogen production. While the authors clearly demonstrate that the nanoporous intermetallic compounds developed in this work are promising for this application, they do not discuss the limitations of the LMD approach in terms of catalyst integration. What is the sample size that can be fabricated, and what are the mechanical properties? Specifically, the electrochemical hydrogen production application requires the fabrication of mechanically robust, hundreds of square centimeter large electrodes. If the authors want to keep the emphasis on high-performance electrochemical hydrogen production, then this needs to be addressed. For the ECSA experiments they used powders prepared by ball milling suggesting that the material is brittle. Thus, spray coating of ball-milled powders on substrates commonly used in this field may be a possible solution as it was recently demonstrated for nanoporous Cu (*Journal of CO2 Utilization* 45, 2021, 101454).

4) On the more fundamental side, I find it hard to believe that the bulk of the nanoporous intermetallic compounds contributes much to the observed activity. The manuscript would strongly benefit from a measurement of the HER related current density for different cathode thicknesses, providing a current density vs overpotential plot where the current density is normalized to both the geometrical footprint of the electrode and the ECSA. This would allow one to differentiate between surface and bulk contributions to the measured current density.

We thank both reviewers for the thorough examination and very helpful comments. The following are our point-by-point replies to the reviewers' comments.

Reviewer #1

Comments:

Presented in this manuscript is the development and application of intermetallic nanoporous alloys for hydrogen evolution. The authors present a unique approach to forming intermetallic alloy nanostructures which are difficult to form through other traditional methods. The demonstration of the mechanism of formation and pore structure stability at elevated temperatures will be a significant addition to the nanoporous metal and dealloying fields. The HER activity of the intermetallic alloys is competitive with a commercial Pt-based catalyst. Find detailed comments and questions below:

Reply: We appreciate the reviewer for the essential summary of our work and for recognizing its significance in the nanoporous metal and dealloying fields. We also appreciate the reviewer's insightful comments, which have made the work much stronger.

1. In the introduction it is mentioned that nanoporous bicontinuous materials have been used extensively for catalysis, however, there are no citations listed. There is a huge body of literature for nanoporous metals for catalysis and electrocatalysis, the authors need to add some of those citations here.

Reply: Thanks for this suggestion. We've added the related references including a few review articles in page 3.

2. In the introduction and results/discussion, the authors need to add more detailed discussion of why intermetallic alloys are expected to be more catalytically active than random/homogeneous alloys. This would add more justification and support for their work.

Reply: According to this suggestion, we have revised our manuscript to include more discussions on the advantages of intermetallic compounds over random solid-solution alloys for electrocatalysis, which are cited below:

Page 3: “Monometallic transition metals (Fe, Co, Ni, Ti, W, Mo, Cu, etc.) show moderate catalytic activity, while alloys of these elements can substantially improve HER catalysis by modifying both active site morphology (geometric effect)⁸⁻¹⁰ and electronic structure (electronic effect)¹¹⁻¹³ to generate desirable catalytic transition states with minimum energy barriers. Compared to random solid-solution alloys, intermetallic compounds with ordered atomic structures and well-defined stoichiometry possess the merits of homogeneous and intensified active sites as well as the capability to form unique crystal structures not commonly demonstrated by random solid-solutions to enhance the geometry effect^{10,14,15}. The combination of electron localization and directional covalent bonding in intermetallic catalysts can also strengthen the electronic effect¹⁴. Moreover, the strong ionic/covalent interactions between the metal constituents are expected to enhance the chemical/electrochemical stability of intermetallic catalysts during catalytic processes.”

Page 11: “The superior HER activity of μ -Co₇Mo₆ over μ -Fe₇Mo₆ could be attributed to the electronic effect that the stronger d-electron interactions between Co and Mo in this hypo–hyper-d-electronic combinations modulate the optimal adsorption/desorption of reaction intermediates^{69,70}.”

3. It is noted that there is a significant difference in formed nanoporous metal surface area depending upon the precursor alloy composition and dealloying parameters. Also, it appears that carbon particles are added to the electrodes, likely to improve conductivity. Are the current densities and activities reported in Figures 5 and 6 normalized by the real active area of the material, or just the geometric surface area of the electrode? Since there are differences in material composition and active area, the activity should be normalized by the real active area as this will indicate whether these differences in activity are just related to the structure of the material or if there is a real kinetic impact of the nanoporous materials. Also, if the active area is determined by double layer capacitance, do the authors anticipate any error in this determinate due to incorporation of carbon particles into the electrodes?

Reply: In our previous manuscript, the electrocatalytic HER performance was evaluated using bulk-form sheet samples, with currents normalized by geometric surface areas. As both Reviewer #1 and Reviewer #2 noted, such tests cannot adequately disclose the intrinsic activity of different samples since there is a distinct difference in pore size and surface area of the nanoporous materials depending upon the precursor alloy compositions and dealloying parameters. Based on these constructive comments and in accordance with the Good Practice Guide for the evaluation of electrocatalysts (*Journal of Power Sources* 451 (2020): 227635), we re-examined the electrocatalytic activities of our materials by testing two types of catalyst

form factors: powder samples and self-supported monolithic sheets. The powder samples were prepared from as-dealloyed nanoporous sheets by mechanical grinding and high-power ultrasonication. 10 mg of nanoporous sample powder was mixed with 2 mg of conductive acetylene black and 50 μL of Nafion solution in 1 mL of ethanol. The mixture was then cast onto a glassy carbon disk electrode (5 mm in diameter) with a catalyst loading of 2 mg cm^{-2} to make thin film electrodes. Such thin film electrodes can reduce the ion diffusion constraints that thick nanoporous sheet electrodes may suffer, and by using the same mixture composition and catalyst loading, allow for a more precise assessment of the intrinsic activity of nanoporous electrocatalysts. In addition to the powder samples, free-standing sheet samples with a size of 1 cm \times 0.5 cm \times 170 μm were also tested as self-supported electrodes for HER to illustrate the potential of directly utilizing mechanically robust monolithic nanoporous intermetallic compounds for electrocatalysis.

(1) Electrochemical properties of the powder samples

Figure R1a-d illustrate the electrochemical HER performance of nanoporous Mo-based intermetallic compounds and the reference materials tested as powder samples cast on a disk electrode: $\mu\text{-Co}_7\text{Mo}_6$, $\mu\text{-Fe}_7\text{Mo}_6$, $\text{Cr}_{50}\text{Mo}_{50}$, Mo, commercial Pt/C, and pure carbon (acetylene black). All nanoporous catalysts were fabricated at the temperature of 973 K. The catalyst loading for all electrodes besides pure carbon is 2 mg cm^{-2} . The pure carbon electrode has a loading of 0.4 mg cm^{-2} , equivalent to the loading of carbon components in the mixture electrodes. Based on the multiple criteria of onset potential (**Figure R1a**), Tafel slope (**Figure R1b, c**), operating potential at 10 $\text{mA cm}^{-2}_{\text{geo}}$ current density (**Figure R1a,c**), and charge transfer resistance (R_{ct}) from electrochemical impedance spectroscopy (EIS) measurements (**Figure R1d**), the catalytic activities exhibit the following trend: Pt/C > $\mu\text{-Co}_7\text{Mo}_6$ > $\mu\text{-Fe}_7\text{Mo}_6$ > Mo > $\text{Cr}_{50}\text{Mo}_{50}$. Pure carbon is essentially inert in the tested potential range. Among the nanoporous electrocatalysts, $\mu\text{-Co}_7\text{Mo}_6$ demonstrates the lowest overpotential of 76 mV at 10 $\text{mA cm}^{-2}_{\text{geo}}$ current density, the lowest Tafel slope of 82 mV dec^{-1} , and the smallest R_{ct} of 5.1 Ω , indicating the highest HER kinetics and catalytic efficiency. The catalytic activity of nanoporous $\mu\text{-Co}_7\text{Mo}_6$ is close, though yet inferior, to that of Pt/C.

To assess the intrinsic activity of the catalysts, the electrochemical surface areas (ECSAs) of the nanoporous electrocatalysts were calculated by measuring their double layer capacitances (C_{dl}), as depicted in **Figure R2**. A pure carbon electrode with a carbon loading of 0.4 mg cm^{-2} was also examined for the capacitive contributions of the carbon component in the mixture electrodes. Compared to the large currents and capacitances of the nanoporous electrodes, those of pure carbon are modest (**Figure R2e,f**). The ECSAs of nanoporous electrocatalysts were determined by subtracting the capacitive contributions from carbon:

$$ECSA = (C_{dl} - C_{dl,carbon})/C_s$$

where C_s is the specific capacitance with a moderate value of $0.040 \text{ mF cm}^{-2}_{ECSA}$. The ECSAs for nanoporous $\mu\text{-Co}_7\text{Mo}_6$, $\mu\text{-Fe}_7\text{Mo}_6$, $\text{Cr}_{50}\text{Mo}_{50}$, and Mo are determined to be 44.6, 54.9, 9.3, and $16.9 \text{ m}^2 \text{ g}^{-1}$, respectively (**Figure R2g**), clearly demonstrating an abundance of active sites in the fine-structured nanoporous intermetallic compounds.

Figure R2h shows the ECSA-normalized HER polarization curves, which suggest an intrinsic catalytic activity trend of the order of $\mu\text{-Co}_7\text{Mo}_6 > \text{Mo} > \text{Cr}_{50}\text{Mo}_{50} > \mu\text{-Fe}_7\text{Mo}_6$. The activity trend for $\mu\text{-Co}_7\text{Mo}_6$, Mo, and $\text{Cr}_{50}\text{Mo}_{50}$ remains unchanged compared to the geometric-surface-area-normalized results. However, nanoporous $\mu\text{-Fe}_7\text{Mo}_6$ exhibits lower intrinsic activity but higher apparent activity (on a geometric surface area basis) than nanoporous Mo and $\text{Cr}_{50}\text{Mo}_{50}$, indicating that the large surface area and associated abundant active sites originating from a small ligament size of nanoporous $\mu\text{-Fe}_7\text{Mo}_6$ contribute to the enhanced catalytic performance. In contrast to $\mu\text{-Fe}_7\text{Mo}_6$, the nanoporous $\mu\text{-Co}_7\text{Mo}_6$ has both the highest intrinsic activity and a fine porous structure for a large surface area, resulting in the highest catalytic activity among the nanoporous electrocatalysts.

The nanoporous $\mu\text{-Co}_7\text{Mo}_6$ fabricated by LMD at various temperatures were also examined as powder samples to determine the effect of pore/ligament size on the electrochemical properties. The three samples prepared at 873 K, 973 K, and 1073 K exhibit ligament diameters of 20.7, 30.8, and 43.1 nm, respectively. As depicted in **Figure R1e**, lower dealloying temperatures and smaller ligament sizes result in larger HER current densities on a geometric surface area basis for a given overpotential, indicating higher catalytic activity. The measured ECSAs for these electrocatalysts are 53.5, 44.6, and $24.1 \text{ m}^2 \text{ g}^{-1}$, respectively (**Figure R3**). Remarkably, the ECSA-normalized HER polarization curves of the three electrocatalysts overlap when the current density is less than $0.037 \text{ mA cm}^{-2}_{ECSA}$ (equivalent to $40 \text{ mA cm}^{-1}_{geo}$ for the 873 K sample), suggesting that they all possess the same intrinsic catalytic activity (**Figure R1f**). Therefore, the smaller-ligament-induced increment in apparent catalytic efficiency is due to a structural effect in which a larger specific surface area provides a larger number of catalytically active sites for electrochemical reactions. This structural effect highlights the significance of the intermetallic effect in realizing the nanoscale porosity structure of Mo-based intermetallic compounds for superior electrochemical performance. Note that when the current density exceeds $40 \text{ mA cm}^{-1}_{geo}$, the 873 K sample's activity decreases and becomes lower than that of the 973 K sample (**Figure R1e,f**). This is probable because the ultrafine porous structure (20.7 nm pore size) of the 873 K sample limits ion/gas transport at high current densities (*Microfluidics and Nanofluidics* 20.7 (2016): 1-13). Under the current testing conditions, the 973 K sample with a pore size of 30.8 nm appears to be the optimal electrocatalyst capable of

balancing the intrinsic reaction kinetics and accessibility of active sites.

(2) Electrochemical properties of the self-supported monolithic sheets

The monolithic nanoporous intermetallic compounds are mechanically robust and can therefore be used as self-supported HER electrodes. **Figure R4a** shows the HER polarization curve of nanoporous $\mu\text{-Co}_7\text{Mo}_6$ sheet electrodes with a dimension of $1\text{ cm} \times 0.5\text{ cm} \times 170\text{ }\mu\text{m}$, in comparison to a Pt/C electrode drop-cast on carbon paper with a similarly high catalyst loading of 24.4 mg cm^{-2} . Due to intensified active sites from the bulk electrode, the nanoporous $\mu\text{-Co}_7\text{Mo}_6$ sheet delivers improved HER performance compared to its powder counterpart, exhibiting a $\sim 0\text{ V}$ onset potential, a 46 mV dec^{-1} Tafel slope, and a 14 mV overpotential at $10\text{ mA cm}^{-2}_{\text{geo}}$ current density (**Figure R4a,b**). The property is among the best performances reported for non-precious Mo-based HER electrocatalysts (**Figure R4c**). Due to the inferior intrinsic activity of nanoporous $\mu\text{-Co}_7\text{Mo}_6$ compared to that of Pt/C, the $\mu\text{-Co}_7\text{Mo}_6$ sheet exhibits larger overpotentials at current densities below $55\text{ mA cm}^{-2}_{\text{geo}}$ than Pt/C (**Figure R4a,b**). In contrast, $\mu\text{-Co}_7\text{Mo}_6$ outperforms Pt/C at high current densities exceeding $55\text{ mA cm}^{-2}_{\text{geo}}$. For instance, an overpotential of 115 mV is required to obtain a current density of $150\text{ mA cm}^{-2}_{\text{geo}}$ for nanoporous $\mu\text{-Co}_7\text{Mo}_6$, which is significantly lower than the 180 mV necessary for Pt/C (**Figure R4b**). While the low-current-density property of the electrocatalysts is mainly controlled by kinetics, at higher current densities the density and accessibility of active sites can dominate the performance, particularly in the porous electrodes. Therefore, the comparison between nanoporous $\mu\text{-Co}_7\text{Mo}_6$ and Pt/C demonstrates that the 3D bicontinuous porous structure facilitates the electrochemical reactions of the intermetallic compounds by readily exposing more active sites, rendering nanoporous $\mu\text{-Co}_7\text{Mo}_6$ particularly suitable for high-current applications. Moreover, unlike Pt/C electrodes, which are prone to exfoliation due to the severe production of H_2 bubbles at high current densities, the robust self-supported $\mu\text{-Co}_7\text{Mo}_6$ sheet with 3D interconnected ligaments has a far better structural stability. Along with excellent electrocatalytic activity, the nanoporous $\mu\text{-Co}_7\text{Mo}_6$ electrodes exhibit excellent durability. After 5000 cycles of accelerated degradation testing by cyclic voltammetry, the nanoporous electrode shows a negligible shift in the HER polarization curves and negligible change in its porous structure (**Figure R4d**). Furthermore, the nanoporous $\mu\text{-Co}_7\text{Mo}_6$ delivers a high current density of $\sim 170\text{ mA cm}^{-2}$ at 150 mV overpotential, without significant current decay over a long-term chronoamperometry test for 48 h (**Figure R4e**).

The above new results and analyses have been included in our revised manuscript (see Pages 10-13).

Figure R1. Electrochemical HER performance of the nanoporous Mo-based intermetallic compounds and the reference materials tested as powder samples cast on a glassy carbon disk electrode. **(a)** iR-corrected HER polarization curves. **(b)** Tafel plots. **(c)** Comparisons of overpotential at $10 \text{ mA cm}^{-2}_{\text{geo}}$ current density and Tafel slopes. **(d)** EIS spectra collected at 200 mV overpotential. The insets are the zoom-in spectra and the electronic circuit for EIS fitting. **(e,f)** iR-corrected HER polarization curves with current normalized by the geometric surface area and the electrochemical active surface area (ECSA), respectively, for np-Co₇Mo₆ fabricated by LMD at the temperatures of 873 K, 973 K, and 1073 K. The catalyst loading is 2 mg cm^{-2} for all samples except for pure carbon.

Figure R2. Cyclic voltammograms at the non-Faradaic potential region for (a) np-Co₇Mo₆, (b) np-Fe₇Mo₆, (c) np-Cr₅₀Mo₅₀, (d) np-Mo, and (e) pure carbon disk electrodes prepared from powder samples. (f) Current density difference ($\Delta j = |j_a - j_c|/2$) at -0.95 V plotted versus the scan rates. The values of double-layer capacitance are given. (g) Electrochemical active surface areas (ECSA) of the various samples. The capacitance contribution from the carbon components in the mixture electrodes was subtracted for calculating the ECSA. (h) iR-corrected HER polarization curves with the current normalized by ECSA.

Figure R3. Cyclic voltammetry curves at the non-Faradaic potential region for np-Co₇Mo₆ fabricated at (a) 873 K, (b) 973 K, and (c) 1073K. The data were collected from disk electrodes prepared from powder samples. (d) Current density difference ($\Delta j = |j_a - j_c|/2$) at -0.95 V plotted versus the scan rates. The values of double-layer capacitance are given. (e) Electrochemical active surface areas (ECSA) of the various samples. The capacitance contribution from the carbon components in the mixture electrodes was subtracted for calculating the ECSA.

Figure R4. HER catalytic performance and durability of the self-supported np- Co_7Mo_6 sheet electrocatalysts (fabricated at 973 K). **(a)** iR-corrected HER polarization curve as compared to a high-load Pt/C catalyst. **(b)** Comparisons of overpotential at 10 and 150 $\text{mA cm}_{\text{geo}}^{-2}$ current density. **(c)** Overpotential at 10 $\text{mA cm}_{\text{geo}}^{-2}$ and Tafel slope of the np- Co_7Mo_6 sheet in comparison with other Mo-based HER electrocatalysts reported in the literature. **(d)** iR-corrected HER polarization curves collected before and after the accelerated degradation test for 5000 cycles. Inset: SEM image of the np- Co_7Mo_6 sheet after the stability measurement. **(e)** Long-term stability measurements for 48 h at 150 mV overpotential.

4. The comparison of HER activity at high current density needs a bit more discussion. Low-current-density performance is controlled by kinetics, but at higher current densities, especially in porous electrodes, ion and gas transport as well as shifts in interfacial pH can dominate the performance. In this regard, can the authors comment on the different in performance of their material and Pt/C at higher current densities?

Reply: We agree with the reviewer that the performance of an electrocatalyst is governed by different mechanisms at low and high current densities. As detailed in our response to Comment #3, we address this question by testing nanoporous intermetallic electrocatalysts as both powder samples and self-supported monolithic sheets. The powder experiments demonstrate that Pt/C has a higher intrinsic catalytic activity than nanoporous Co_7Mo_6 (**Figure R1a,b**). Consistently, the np- Co_7Mo_6 exhibits larger overpotentials than Pt/C for self-supported sheet samples at current densities $< 55 \text{ mA cm}^{-2}_{\text{geo}}$ (**Figure R4a,b**). While at high current densities where ion/gas transport and the availability and accessibility of active sites influence performance, the np- Co_7Mo_6 sheet outperforms Pt/C. For instance, an overpotential of 115 mV is required to obtain a current density of $150 \text{ mA cm}^{-2}_{\text{geo}}$ for np- Co_7Mo_6 , which is significantly lower than the 180 mV necessary for Pt/C. (**Figure R4a,b**). The comparison between nanoporous $\mu\text{-Co}_7\text{Mo}_6$ and Pt/C demonstrates that the 3D bicontinuous porous structure facilitates the electrochemical reactions of the intermetallic compounds by readily exposing more active sites, rendering the nanoporous $\mu\text{-Co}_7\text{Mo}_6$ particularly suitable for high-current applications. We've added more discussion on this aspect in our revised manuscript:

Page 12: *“Due to the inferior intrinsic activity of nanoporous $\mu\text{-Co}_7\text{Mo}_6$ compared to that of Pt/C, the $\mu\text{-Co}_7\text{Mo}_6$ sheet exhibits larger overpotentials at current densities below $55 \text{ mA cm}^{-2}_{\text{geo}}$ than Pt/C (Fig. 6a). In contrast, $\mu\text{-Co}_7\text{Mo}_6$ outperforms Pt/C at high current densities exceeding $55 \text{ mA cm}^{-2}_{\text{geo}}$. For instance, an overpotential of 115 mV is required to obtain a current density of $150 \text{ mA cm}^{-2}_{\text{geo}}$ for nanoporous $\mu\text{-Co}_7\text{Mo}_6$, which is significantly lower than the 180 mV necessary for Pt/C (Fig. 6b). While the low-current-density property of the electrocatalysts is mainly controlled by kinetics, at higher current densities the density and accessibility of active sites can dominate the performance, particularly in the porous electrodes. Therefore, the comparison between nanoporous $\mu\text{-Co}_7\text{Mo}_6$ and Pt/C demonstrates that the 3D bicontinuous porous structure facilitates the electrochemical reactions of the intermetallic compounds by readily exposing more active sites, rendering the nanoporous $\mu\text{-Co}_7\text{Mo}_6$ particularly suitable for high-current applications. Moreover, unlike Pt/C electrodes, which are prone to exfoliation due to the severe production of H_2 bubbles at high current densities, the robust self-supported $\mu\text{-Co}_7\text{Mo}_6$ sheet with 3D interconnected ligaments has a far better structural stability.”*

5. In Figure 1g and 1h, it appears that the nanoporous material is polycrystalline with nanosized grains. In traditional dealloying, the dealloyed material retains the crystal and grain structure of the parent alloy. Is it different here? It would add to the manuscript if the authors can add some discussion here.

Reply: We appreciate the reviewer's insight. Indeed, for the most extensively studied

Au-Ag alloys, the dealloyed nanoporous gold retains the grain structure of the parent alloy and the ligament feature sizes are typically orders of magnitude smaller than grain sizes. This remarkable behavior originates from the uniqueness of the Au-Ag alloy system, which has continuous solid solubility and very small differences in atomic volumes between Au and Ag, enabling a crystallographically epitaxial relationship between Au and Au-Ag alloy and an essentially coherent transformation from master alloy to nanoporous product structure. However, most dealloying systems are not as ideal as the prototypical nanoporous gold. For alloys such as metallic glasses and intermetallic compounds that cannot establish such epitaxy or undergo phase transformation, the dealloying often yields nanoporous materials with an abundance of grain boundaries inside the porous network. As a representative example, Paschalidou et al. reported that nanoporous gold formed by dealloying an amorphous alloy is composed of polycrystalline ligaments, as opposed to single crystals obtained from crystalline alloys (*Acta Materialia* 119 (2016) 177-183). Similarly, nanoporous Si produced by liquid metal dealloying of Mg₂Si alloy (*Nano Lett.* 2014, 14, 4505–4510) and nanoporous Ni by vapor phase dealloying of Ni₂Zn₁₁ alloy (*Acta Materialia* 2019, 163, 161-172) have comparable grain size and ligament sizes. For the present study, the LMD is fulfilled by the phase transformation from fcc-Ni₇₀(Co_{0.55}Mo_{0.45})₃₀ to trigonal μ -Co₇Mo₆ with distinctly different crystal structure and lattice parameters (**Table R1**). Consequently, the development of μ -Co₇Mo₆ ligaments during the dealloying of fcc-Ni₇₀(Co_{0.55}Mo_{0.45})₃₀ cannot remain coherent and epitaxial, but is more likely through an interfacial heterogeneous nucleation and growth process, resulting in the formation of polycrystalline ligaments with nanosized grains.

These discussions were incorporated to the revised manuscript in Page 6.

Table R1. Unit cell parameters of fcc-Ni₇₀(Co_{0.55}Mo_{0.45})₃₀ and μ -Co₇Mo₆

	Crystal System	Space group	a (Å)	b (Å)	c (Å)
Ni ₇₀ (Co _{0.55} Mo _{0.45}) ₃₀	Cubic	Fm-3m	3.585	3.585	3.585
μ -Co ₇ Mo ₆	Trigonal	R-3m	4.762	4.762	25.617

6. The CoMo material is dealloyed at different temperatures to evolve different nanostructures. Is the composition the same for all nanoporous metals, regardless of temperature used to dealloy?

Reply: We appreciate the reviewer's comment. The chemical compositions of the nanoporous materials were determined using SEM-EDS. The results are provided in **Supplementary Table 1** (cited below as **Table R2**), showing that the nanoporous

Co₇Mo₆ dealloyed at 873 K, 973 K and 1073 K have nearly identical compositions. This is further confirmed by the fact that these three electrocatalysts have the same intrinsic catalytic activity (as evidenced by the ECSA-normalized HER polarization curves), indicating that they are composed of the same material.

Table R2. Chemical compositions (at.%) of the precursor alloys and the dealloyed np-M-Mo (M = Co, Fe, Cr) samples prepared at different temperatures by LMD.

	Sample	Ni	Co	Fe	Cr	Mo	Mg
Before LMD	Ni ₇₀ Mo ₃₀	71.20	-	-	-	28.80	-
	Ni ₇₀ Fe ₃₀	69.58	-	30.42	-	-	-
	Ni ₇₀ (Co _{0.55} Mo _{0.45}) ₃₀	69.30	16.80	-	-	13.90	-
	Ni ₇₀ (Fe _{0.58} Mo _{0.42}) ₃₀	69.90	-	17.25	-	12.85	-
	Ni ₇₀ (Cr _{0.50} Mo _{0.50}) ₃₀	70.27	-	-	15.05	14.68	-
After LMD	np-Mo (973 K)	0.56	-	-	-	99.42	0.01
	np-Co ₇ Mo ₆ (873 K)	7.69	46.01	-	-	46.30	0
	np-Co ₇ Mo ₆ (973 K)	4.30	50.20	-	-	44.25	1.25
	np-Co ₇ Mo ₆ (1073 K)	4.63	47.67	-	-	46.86	0.84
	np-Fe ₇ Mo ₆ (973 K)	2.12	-	54.42	-	43.11	0.34
	np-Cr ₅₀ Mo ₅₀ (973 K)	0.38	-	-	49.68	49.92	0.02
	np-Cr ₅₀ Mo ₅₀ (1173 K)	0.74	-	-	49.89	49.38	0

Reviewer #2

Comments:

The authors report on the fabrication of monolithic nanoporous intermetallic compounds via liquid metal dealloying (LMD) and their performance for electrocatalytic hydrogen production. Using starting alloys that result in the formation of intermetallic compounds during LMD is a clever idea, and experiments are well designed and beautifully executed. Using experiments and theory, the authors convincingly demonstrate that the ultra-fine feature size observed for intermetallic μ -Co₇Mo₆ and μ -Fe₇Mo₆ is a consequence of the suppressed Mo atomic diffusivity. The reviewer believes that these experiments alone justify publication in Nature Communications. However, the comments listed below should be considered before

acceptance of the manuscript.

Reply: We thank the reviewer for the positive evaluation of our work and the very constructive comments. Accordingly, we have carried out new experiments and additional analyses to improve our work. The detailed can be found below.

1) I could not find the information about the size of the samples they produced and used for the electrochemical experiments (only the thickness was provided)

Reply: In accordance with this comment, we have provided optical pictures of the samples at various stages of the fabrication procedure. As shown in **Figure R5a**, a 200 μm -thick $\text{Ni}_{70}(\text{Co}_{0.55}\text{Mo}_{0.45})_{30}$ alloy sheet with a size of 2 cm \times 1 cm was converted into a free-standing nanoporous Co_7Mo_6 sheet with a well-preserved lateral dimension of 2 cm \times 1 cm. As determined by SEM, the thickness of the sheet decreased from 200 μm to 170 μm due to volume contraction during LMD. For electrochemical measurements employing self-supported sheet samples, the sheet electrodes have a size of 1 cm \times 0.5 cm \times 170 μm as shown in **Figure R5b-d**. The sample size information and optical pictures have been added to our revised manuscript, including **Figure 1c** and **Supplementary Figure 21**.

Figure R5. (a) Optical images of a 200 μm -thick $\text{Ni}_{70}(\text{Co}_{0.55}\text{Mo}_{0.45})_{30}$ alloy sheet with a size of 2 cm \times 1 cm, the $\text{np-Co}_7\text{Mo}_6/\text{Mg}$ composite obtained after LMD, and the $\text{np-Co}_7\text{Mo}_6$ obtained after further Mg removal. (b-e) Optical images of three self-supported $\text{np-Co}_7\text{Mo}_6$ sheet electrodes with a dimension of 1 cm \times 0.5 cm \times 170 μm and a high-load Pt/C electrode.

2) No information about the mechanical properties is provided. For the ECSA

experiments the authors used powders prepared by ball milling suggesting that the material is brittle. It is not necessary to add mechanical tests, but at least a qualitative description of the mechanical behavior and handling properties should be added.

Reply: We thank the reviewer for this comment. Yes, the nanoporous intermetallic compounds are brittle due to the nanoporous structure and the inherent brittleness of intermetallic compounds. In spite of this, the monolithic nanoporous intermetallic compounds possess considerable mechanical robustness: they retain structural integrity during LMD and Mg removal (**Figure R5a**) and can be handled without breaking. We provided a movie (**Video S1**) and snapshots (**Figure R6**) to show the handling property and the brittle feature of nanoporous Co₇Mo₆.

Inspired by the Reviewer's comment, we complemented mechanical tests for a better understanding of the material. As the nanoporous intermetallic compounds can inherit the macroscopic shape of the alloy precursor, we prepared the np-Co₇Mo₆ sample for tensile testing by dealloying a dogbone-shaped precursor alloy (**Figure R7a**). The resultant np-Co₇Mo₆ sample has a gauge length of 3.8 mm, a width of 1.5 mm and a thickness of 170 μm. The tensile tests were conducted at a strain rate of $2 \times 10^{-3} \text{ s}^{-1}$ using the high precision Shimadzu EZ-SX instrument equipped with a 1 N load cell. As shown in **Figure R7b**, the tensile stress-strain curves of three np-Co₇Mo₆ samples are linear elastic to sample fracture with no detectable macro-scale plastic yield behaviour, which is characteristic of a brittle material. The fracture strength ranges from 1.8 to 2.2 MPa and the average value is ~2.0 MPa. For comparison, the fracture strengths of nanoporous Au (Acta Materialia 129 (2017) 251-258) and nanoporous Ni (Acta Materialia 166 (2019) 402-412) in tension were 10~20 MPa and ~7.9 MPa, respectively. The lower fracture strength of np-Co₇Mo₆ is most likely caused by the abundant pre-existing large grain boundary cracks formed during dealloying and/or chemical etching for removing Mg, which can be clearly seen in **Figure R7c** (indicated by red arrows). The fracture surface of np-Co₇Mo₆ after tensile test was characterized by SEM (**Figure R7c-e**), which shows two distinct fracture zones: I, transgranular rough surfaces, and II, the flat region from intergranular brittle fracture. The magnified images of the ruptured ligaments reveal brittle fracture without necking (**Figure R7d**), which is consistent with the inherent brittleness of the intermetallic compounds.

These new results have been included in the revised manuscript as **Supplementary Figure 4** and **Supplementary Figure 5**.

Figure R6. Snapshots of a bending test showing the handling property and brittleness of free-standing np-Co₇Mo₆ sheet.

Figure R7. Mechanical properties of the np-Co₇Mo₆. **(a)** Optical image of the precursor

alloy and the np-Co₇Mo₆ sample for tensile test. The precursor alloy is cut into the dogbone shape with a gauge length of 3.8 mm, a width of 1.5 mm and a thickness of 200 μm. The macroscopic shape is well preserved after LMD and Mg removal with only slight thickness change from 200 μm to 170 μm due to volume contraction. **(b)** Tensile stress-strain curves of three np-Co₇Mo₆ samples. The fracture strength ranges from 1.8 to 2.2 MPa and the average value is ~2.0 MPa. **(c)** The fracture surface of np-Co₇Mo₆ after tensile test. The surface shows two different fracture zones: I, transgranular rough surfaces, and II, the flat region from intergranular brittle fracture (marked by red dotted lines). Lamellar cracks formed during dealloying (marked by red arrows) can also be seen. **(d,e)** Zoom-in views of the fracture surfaces from region I and II, respectively.

3) My main critic is the apparent focus on high-performance electrochemical hydrogen production. While the authors clearly demonstrate that the nanoporous intermetallic compounds developed in this work are promising for this application, they do not discuss the limitations of the LMD approach in terms of catalyst integration. What is the sample size that can be fabricated, and what are the mechanical properties? Specifically, the electrochemical hydrogen production application requires the fabrication of mechanically robust, hundreds of square centimeter large electrodes. If the authors want to keep the emphasis on high-performance electrochemical hydrogen production, then this needs to be addressed. For the ECSA experiments they used powders prepared by ball milling suggesting that the material is brittle. Thus, spray coating of ball-milled powders on substrates commonly used in this field may be a possible solution as it was recently demonstrated for nanoporous Cu (Journal of CO₂ Utilization 45, 2021, 101454).

Reply: We thank the reviewer for the insightful and constructive comments. Inspired by this remark, we tested electrochemical performance of our nanoporous intermetallic compounds with two different types of catalyst form factors: powder samples and self-supported monolithic sheets. The powder samples were generated from as-dealloyed sheet samples using a combination of mechanical grinding and high-power ultrasonication (**Figure R8**). From the powder samples, thin film electrodes were fabricated, which can largely eliminate the ion diffusion limitations associated with thick nanoporous sheet electrodes and allow for a more accurate evaluation of the intrinsic activity of nanoporous electrocatalysts. The electrochemical performance of the powder samples also reveals how the electrocatalysts would function if they were sprayed onto substrates to produce electrodes. The powder sample testing demonstrates that the nanoporous μ-Co₇Mo₆ has both high intrinsic activity and a fine porous structure for a large surface area, enabling a superb catalytic performance (**Figure R1**).

Among the nanoporous electrocatalysts, the μ -Co₇Mo₆ delivers the lowest overpotential of 76 mV at 10 mA cm⁻²_{geo} current density, the lowest Tafel slope of 82 mV dec⁻¹, and the smallest R_{ct} of 5.1 Ω , indicating the highest HER kinetics and catalytic efficiency. The catalytic activity of np-Co₇Mo₆ is close, though yet inferior, to that of Pt/C. In addition to the powder samples, free-standing sheet samples with a size of 1 cm \times 0.5 cm \times 170 μ m were also tested as self-supported electrodes for HER to illustrate the potential of directly employing mechanically robust monolithic nanoporous intermetallic compounds for electrocatalysis. The results show that the self-supported nanoporous μ -Co₇Mo₆ sheet delivers improved HER performance compared to its powder counterpart due to intensified active sites, exhibiting a \sim 0 V onset potential, a 46 mV dec⁻¹ Tafel slope, and a 14 mV overpotential at 10 mA cm⁻²_{geo} current density (**Figure R4**). Particularly, the μ -Co₇Mo₆ outperforms Pt/C at high current densities exceeding 55 mA cm⁻²_{geo}, which demonstrates that the nanoporous μ -Co₇Mo₆ is particularly suitable for high-current applications. More detailed results and discussions can be found in our response to Comment #3 from Reviewer #1.

In the revised manuscript, we also included critical discussions on the limitations and challenges associated with the catalytic integration of our nanoporous intermetallic compounds, which are cited below.

Page 13: *“It is necessary to note that despite the enhanced catalytic efficiency of self-supported nanoporous μ -Co₇Mo₆, the sheet electrodes have a much lower material utilization efficiency than powder samples (Supplementary Fig. 23) because only the near surface layers of the thick electrodes may contribute to electrochemical reactions, particularly at a high current density. High material efficiency would necessitate further optimization of the sheet thickness and the nanoporous structure. In addition, advanced catalyst integration techniques must be developed in order to implement the intrinsically brittle nanoporous μ -Co₇Mo₆ sheet as large-sized electrodes in commercial electrolyzers. By contrast, the powder form of the nanoporous intermetallic electrocatalysts, which can be spray-coated on substrates to make scalable electrodes with high material efficiency⁷², might be readily applicable for practical hydrogen production applications.”*

Figure R8. Optical images of a free-standing $\text{np-Co}_7\text{Mo}_6$ sheet immersed in ethanol: (a) original state; (b) after ultrasonic treatment for 15 min at the low power of 65 W; (c) after ultrasonic treatment for an additional 15 min at the high power of 130 W; (d) after sedimentation. The free-standing $\text{np-Co}_7\text{Mo}_6$ sheet remains robust during the ultrasonication, but high-power ultrasonication can peel the sheet layer by layer to make powder samples. Therefore, a combination of mechanical grinding with a mortar and pestle and high-power ultrasonication at 130 W can be used to prepare powder samples from the as-dealloyed $\text{np-Co}_7\text{Mo}_6$ sheets.

4) On the more fundamental side, I find it hard to believe that the bulk of the nanoporous intermetallic compounds contributes much to the observed activity. The manuscript would strongly benefit from a measurement of the HER related current density for different cathode thicknesses, providing a current density vs overpotential plot where the current density is normalized to both the geometrical footprint of the electrode and the ECSA. This would allow one to differentiate between surface and bulk contributions to the measured current density.

Reply: We agree with the reviewer that the bulk of thick nanoporous intermetallic compound electrodes could contribute little to electrocatalytic reactions at high current density due to ion and gas transport limitations. In our revised manuscript, we further tested the electrocatalytic activities by evaluating powder samples that were mixed with carbon and cast as thin films on a disk electrode. Such thin film electrodes can reduce the ion diffusion constraints that thick nanoporous sheet electrodes may suffer, and allow for a more precise assessment of the intrinsic activity of nanoporous electrocatalysts by using the same mixture composition and catalyst loading. We presented and analyzed the HER polarization curves (i.e., the current density vs overpotential plot) with the current normalized to both the geometrical area of the electrode (which represents the apparent catalytic activity) and the ECSA (which reveals the intrinsic activity) (**Figure R9**). According to **Figure R9a**, the apparent catalytic activities of the intermetallic compounds and the reference materials exhibit

the following trend: Pt/C > μ -Co₇Mo₆ > μ -Fe₇Mo₆ > Mo > Cr₅₀Mo₅₀. Pure carbon is essentially inert in the tested potential range. While the ECSA-normalized results in **Figure R9b** suggest an intrinsic catalytic activity trend of the order of μ -Co₇Mo₆ > Mo > Cr₅₀Mo₅₀ > μ -Fe₇Mo₆. In both analyses, the activity trend for μ -Co₇Mo₆, Mo, and Cr₅₀Mo₅₀ remains unchanged. However, nanoporous μ -Fe₇Mo₆ exhibits lower intrinsic activity but higher apparent activity than nanoporous Mo and Cr₅₀Mo₅₀, indicating that the large surface area and associated abundant active sites originating from a small ligament size of nanoporous μ -Fe₇Mo₆ contribute to the enhanced catalytic performance. In contrast to μ -Fe₇Mo₆, the nanoporous μ -Co₇Mo₆ has both the highest intrinsic activity and a fine porous structure for a large surface area, resulting in the highest catalytic activity among the nanoporous electrocatalysts. The superior HER activity of μ -Co₇Mo₆ over μ -Fe₇Mo₆ could be attributed to the electronic effect that the stronger *d*-electron interactions between Co and Mo in this hypo–hyper-*d*-electronic combinations modulate the optimal adsorption/desorption of reaction intermediates (*Mater. Chem. Phys.* 22, 1-26 (1989); *Electrochim. Acta* 45, 4085-4099 (2000)).

For nanoporous μ -Co₇Mo₆ fabricated by LMD at various temperatures (873 K, 973 K, and 1073 K), the results in **Figure R9c** show that lower dealloying temperatures and smaller ligament sizes result in larger HER current densities on a geometric surface area basis for a given overpotential, indicating higher apparent catalytic activity. Remarkably, the ESCA-normalized HER polarization curves of the three electrocatalysts overlap when the current density is less than 0.037 mA cm⁻²_{ECSA} (equivalent to 40 mA cm⁻¹_{geo} for the 873 K sample), suggesting that they all possess the same intrinsic catalytic activity (**Figure R9d**). Therefore, the smaller-ligament-induced increment in apparent catalytic efficiency is due to a structural effect in which a larger specific surface area provides a larger number of catalytically active sites for electrochemical reactions. This structural effect highlights the significance of the intermetallic effect in realizing the nanoscale porosity structure of Mo-based intermetallic compounds for superior electrochemical performance. Note that when the current density exceeds 40 mA cm⁻¹_{geo}, the 873 K sample's activity decreases and becomes lower than that of the 973 K sample (**Figure R9d**). This is probable because the ultrafine porous structure (20.7 nm pore size) of the 873 K sample limits ion/gas transport at high current densities (*Microfluid. Nanofluid.* 20, 1-13 (2016)). Under the current testing conditions, the 973 K sample with a pore size of 30.8 nm appears to be the optimal electrocatalyst capable of balancing the intrinsic reaction kinetics and accessibility of active sites.

In addition to the more thorough electrochemical testing using both powder and self-supported monolithic sheets in our revised manuscript, the limits of nanoporous sheet electrodes in terms of materials utilization efficiency were discussed (**Figure R10**):

Page 13: “It is necessary to note that despite the enhanced catalytic efficiency of self-supported nanoporous $\mu\text{-Co}_7\text{Mo}_6$, the sheet electrodes have a much lower material utilization efficiency than powder samples (Supplementary Fig. 23) because only the near surface layers of the thick electrodes may contribute to electrochemical reactions, particularly at a high current density. High material efficiency would necessitate further optimization of the sheet thickness and the nanoporous structure.”

Figure R9. Electrochemical HER performance of the powder samples. **(a,b)** iR-corrected HER polarization curves with current normalized by the geometric surface area and the electrochemical active surface area (ECSA), respectively, for the nanoporous Mo-based intermetallic compounds and the reference materials. **(c,d)** iR-corrected HER polarization curves with current normalized by the geometric surface area and the electrochemical active surface area (ECSA), respectively, for np-Co₇Mo₆ fabricated by LMD at the temperatures of 873 K, 973 K, and 1073 K. The catalyst loading is 2 mg cm⁻² for all samples except for pure carbon (which has a loading of 0.4 mg cm⁻², equivalent to the loading of carbon components in the mixture electrodes.)

Figure R10. iR-corrected HER polarization curves of np-Co₇Mo₆ sheet and powder electrodes with the current normalized by the geometric area and the mass of electrocatalyst, showing the higher materials utilization efficiency of the powder electrode than the self-supported sheet electrode.

REVIEWERS' COMMENTS

Reviewer #1 (Remarks to the Author):

My concerns have been addressed.

Reviewer #2 (Remarks to the Author):

The authors have addressed all remaining questions and comments, and added new experimental data to support their response. No further changes are necessary, and the manuscript can now be published as is.

Reviewer #1

My concerns have been addressed.

Reply: We thank the reviewer for supporting the publication of our paper.

Reviewer #2

The authors have addressed all remaining questions and comments, and added new experimental data to support their response. No further changes are necessary, and the manuscript can now be published as is.

Reply: We thank the reviewer for supporting the publication of our paper in Nature Communications.